# Non-canonical IL-22 receptor signaling remodels the oral mucosal barrier during *Candida albicans* immunosurveillance

Nicolas Millet [1,2,9], Jinendiran Sekar [1,2], Norma V. Solis[1,2], Jian Miao [1,2], Antoine Millet[2,3], Felix E. Y. Aggor[4], Asia Wildeman[1,2], Melissa E. Cook [4], Amirhossein Davari [5], Brian M. Peters [6], Michail S. Lionakis [7], Sarah L. Gaffen [4], Nicholas Jendzjowsky [2,3,8], Scott G. Filler [1,2,8] & Marc Swidergall [1,2,8] ✉

Mucosal barrier integrity is vital for homeostasis with commensal organisms while preventing pathogen invasion. Here we show that fungal-induced immunosurveillance enhances resistance to fungal outgrowth and tissue invasion by remodeling the oral mucosal epithelial barrier in mouse models of adult and neonatal *Candida albicans* colonization. Epithelial subset expansion and tissue remodeling are dependent on interleukin-22 and signal transducer and activator of transcription 3 signaling, through a non-canonical receptor complex composed of glycoprotein 130 coupled with the interleukin-22 receptor subunit alpha-1 and the interleukin-10 receptor subunit beta. Epithelial proliferation enhanced antifungal host defenses through the upregulation of antimicrobial peptide expression. Immunosurveillance-induced epithelial remodeling is restricted to the oral mucosa, whereas barrier architecture is reset once fungal-specific immunity developed. Collectively, these findings identify fungal-induced transient mucosal remodeling as a critical determinant of resistance to mucosal fungal infection during early stages of microbial colonization.

Oral mucosal tissues are continuously exposed to food, airborne antigens, and commensal microbes, including fungi[1]. The finely tuned implementation of innate and adaptive immune responses enables the host to maintain homeostasis with commensal species and neutralize invading organisms[2–4]. As a central constituent of the mycobiome, the fungus *Candida albicans* colonizes the oral and gastrointestinal (GI) mucosa of up to 75% of healthy individuals[5]. Settings of local or systemic immunosuppression result in oropharyngeal candidiasis (OPC)[6].

Since, in healthy individuals *C. albicans* causes no harm, fungal colonization appears to be evolutionarily selected for appropriate metabolic function and immune priming[7–10]. While *C. albicans* has been extensively studied as a pathogen[11–14], the primary lifestyle of this fungus in the oral cavity is as a commensal[15–17]. The natural diversity in *C. albicans* influences the outcome of the interaction between the fungus and the host[16] implying that the involvement of specific immune pathways to the host defense against *C. albicans* is modulated

[1]Division of Infectious Diseases, Harbor-UCLA Medical Center, Torrance, CA, USA. [2]The Lundquist Institute for Biomedical Innovation at Harbor-UCLA Medical Center, Torrance, CA, USA. [3]Division of Respiratory and Critical Care Medicine and Physiology, Harbor-UCLA Medical Center, Torrance, CA, USA. [4]University of Pittsburgh, Division of Rheumatology and Clinical Immunology, Pittsburgh, PA, USA. [5]Graduate Program in Pharmaceutical Sciences, College of Graduate Health Sciences, University of Tennessee Health Science Center, Memphis, TN, USA. [6]Department of Clinical Pharmacy and Translational Science, College of Pharmacy, University of Tennessee Health Science Center, Memphis, TN, USA. [7]Fungal Pathogenesis Section, Laboratory of Clinical Immunology and Microbiology (LCIM), National Institute of Allergy and Infectious Diseases (NIAID), Bethesda, MD, USA. [8]David Geffen School of Medicine at UCLA, Los Angeles, CA, USA. [9]Present address: Sorbonne Université, INSERM, Centre de Recherche Saint-Antoine (CRSA), Paris, France. ✉e-mail: mswidergall@lundquist.org

in a strain-dependent manner. In mouse models of OPC, *C. albicans* strains can be grouped into two categories: clearance-biased (CB) and persistence-biased (PB)[16]. While CB *C. albicans* strains induce a strong pro-inflammatory response that leads to rapid clearance in the oral mucosa (acute OPC), PB *C. albicans* strains instead trigger a tempered inflammatory response that permits long-term fungal colonization, thus mimicking commensal colonization in humans. Still, the regulation of mucosal homeostasis during persistent fungal colonization remains poorly understood.

The epithelial architecture of mucosal surfaces such as the oral cavity is crucial for its host defensive function[18–20], providing structural immunity by initiating and coordinating immune responses[21]. However, relatively little is known about how this is coordinated in the oral mucosa. Furthermore, the epithelium balances a multiplicity of roles in early life[22], while the acquired microbiome contributes to the development of immunity in newborns. Following exposure, the mucosal immune system of neonates undergoes successive, non-redundant phases that support the developmental needs of the infant to establish immune homeostasis[23]. While tissue remodeling has been associated with pathological features post-injury or disease[24,25], commensal organisms may induce changes in barrier structures to promote homeostasis[26].

Here, we studied mucosal immune responses and tissue homeostasis in mouse models of persistent *Candida* oral colonization at distinct ranges of age. We show that immunosurveillance-induced epithelial expansion and remodeling mediates resistance against fungal outgrowth during the onset of fungal colonization. IL-22, a critical cytokine in epithelial homeostasis and host defense at mucosal surfaces, has long been associated with signaling through a well-characterized receptor complex[27]. Traditionally, IL-22 exerts its protective effects through binding to the IL22RA1-IL10RB receptor complex. However, our study reveals that IL-22 recognition and signaling extend beyond this canonical pathway, expanding the current understanding of its biological roles. Oral mucosal remodeling required IL-22-mediated gp130 activation in non-canonical cytokine receptor complexes with IL-22RA1 and IL-10RB to amplify antimicrobial peptide (AMP) expression. IL-22 mediated oral epithelial remodeling was transitory and a subsequent mucosal remodeling event in later stages of colonization required *Candida*-specific immunity. Finally, fungal-induced mucosal remodeling prevents tissue invasion in a mouse model of neonatal colonization. These findings provide insight into the molecular mechanisms of IL-22-mediated signaling, identify a pathway that supports antifungal immunity during fungal persistence, and expand our understanding of microbe-induced epithelial remodeling and homeostasis through a non-canonical receptor complex.

## Results

### Persistent *C. albicans* colonization induces epithelial remodeling

In a mouse model of OPC[28], acute infection with a prototypic CB *C. albicans* strain (SC5314) leads to its rapid clearance from the oral cavity, whereas a PB *Candida* strain (CA101) persists in the oral mucosa (Supplementary Fig. S1)[15]. A similar trend was observed in the GI tract following oral infection (Supplementary Fig. S1). However, oral colonization alone does not result in systemic dissemination in the absence of immunosuppression (Supplementary Fig. S1). Several host cell types respond to *C. albicans* encounter in the oral cavity. To develop a single cell transcriptome profile of the oral mucosa, we evaluated gene expression in tissues of PB colonized mice and from mice after pathogenic *Candida* clearance (Fig. 1a) representing distinct post-infection host states, namely immune and barrier responses post-fungal clearance (CB) and one reflecting ongoing *Candida* presence and interaction (PB). Our analysis identified 18 distinct cell subpopulations, which were present in both, PB colonized mucosa and mucosal tissue after CB clearance (Fig. 1b). The identified

subpopulations expressed cell-type specific marker genes (Supplementary Fig. S2) consistent with classical, well-established markers for each respective cell population. The scRNA-sequencing analysis revealed that the epithelial proportions increased during persistent fungal colonization (Fig. 1c). Accordingly, epithelial cells from colonized mice had higher expression of proliferation marker genes (*Mki67*, *Cenpf*, *Cenpa*) (Fig. 1d). Tissue histology during the onset of PB *Candida* colonization revealed that, in contrast to CB-infected mice, the epithelial layer expanded (Fig. 1e) and basal epithelial cell proliferation increased indicated by Ki67 staining (Fig. 1f, g). Next, we performed RNA sequencing of epithelial-enriched mucosal tissue (Fig. 2a). Using Gene Set Enrichment Analysis (GSEA), we found that genes involved in keratinization and keratinocyte differentiation were significantly enriched in fungal colonized epithelial tissues (Fig. 2b). Similarly, genes for keratinization were enriched in epithelial cells from colonized mice in the scRNA-sequencing data set (Supplementary Fig. S3). Keratins influence the epithelial architecture by determining cell compartmentalization and differentiation[29]. Several keratins, including keratin 14 (K14), were highly expressed in the mucosa of colonized mice (Fig. 2c). Basal epithelial cells express K14 in all regions, while the suprabasal epithelial layer (SEL) expresses K13[30]. Consistent with these findings the K14 epithelial layer expanded during persistent colonization, while the epithelium had similar architecture after CB encounter and clearance compared to sham infection (Fig. 2d–f). Notably, the K13 layer distribution and thickness remained unchanged during fungal persistence (Supplementary Fig. S4). Time course studies revealed that K14 epithelial expansion occurs after 8 days of fungal colonization (Fig. 2g, Supplementary Fig. S5). Similar oral epithelial expansion was observed by diverse *C. albicans* clinical isolates from vaginal, catheter-associated, and oral environments (Supplementary Fig. S6), indicating that *Candida* persistence induces conserved epithelial remodeling responses within the oral mucosa. Collectively, these data show that persistent fungal colonization induces distinct epithelial subset expansion; thus, indicating that ongoing *Candida* presence influences the oral mucosal architecture.

### CD4$^+$ T cells are a major source of IL-22 during persistent colonization

T cells play vital roles in the mucosal antifungal immunity[31,32], while some CD4$^+$ T cells reside in the oral mucosa of healthy individuals[20]. In fact, T cells can promote stromal cell proliferation through secretion of cytokines[33–35]. Thus, we compared protein expression kinetics of various T helper (Th) cell-associated cytokines and chemokines during persistent *C. albicans* colonization and acute OPC. Acute infection with the CB *C. albicans* strain led to early IL-1β, TNFα, and IFNγ induction, whereas the PB *Candida* strain colonized the oral mucosa without inducing canonical pro-inflammatory cytokines (IL-1β, TNFα, IFNγ) and T cell-polarizing cytokines (IL-1β, IL-6, IL-23) (Fig. 3A; Supplementary Fig. S7). In acute OPC, IL-17 and IL-22 are expressed by type 17 cells with similar kinetics[11,36–38]. In contrast, our data show that IL-22 and IL-17 are differentially expressed during persistent colonization (Fig. 3a, b). Mice infected with a CB strain induced IL-17A and IL-22 similarly followed by rapid decline after fungal clearance. However, in PB-colonized mice, mucosal IL-22 levels remained high, whereas IL-17A levels declined to low levels at the onset of colonization (Fig. 3b). Thus, PB colonization induces limited inflammation characterized primarily by a type 17 response (IL-17/IL-22), rather than broad pro-inflammatory activation as seen during the early phase of CB infection. IL-22 neutralization after PB *C. albicans* colonization (Fig. 3c) resulted in fungal proliferation (Fig. 3d), while antibody neutralization of IL-17A did not alter the fungal burden in colonized mice. This suggested that IL-22 may have a prominent role in controlling fungal outgrowth after mucosal colonization, while IL-17 signaling may be more critical at earlier stages of infection[16], without immediately affecting fungal burden at the onset of oral colonization. Next, we determined the

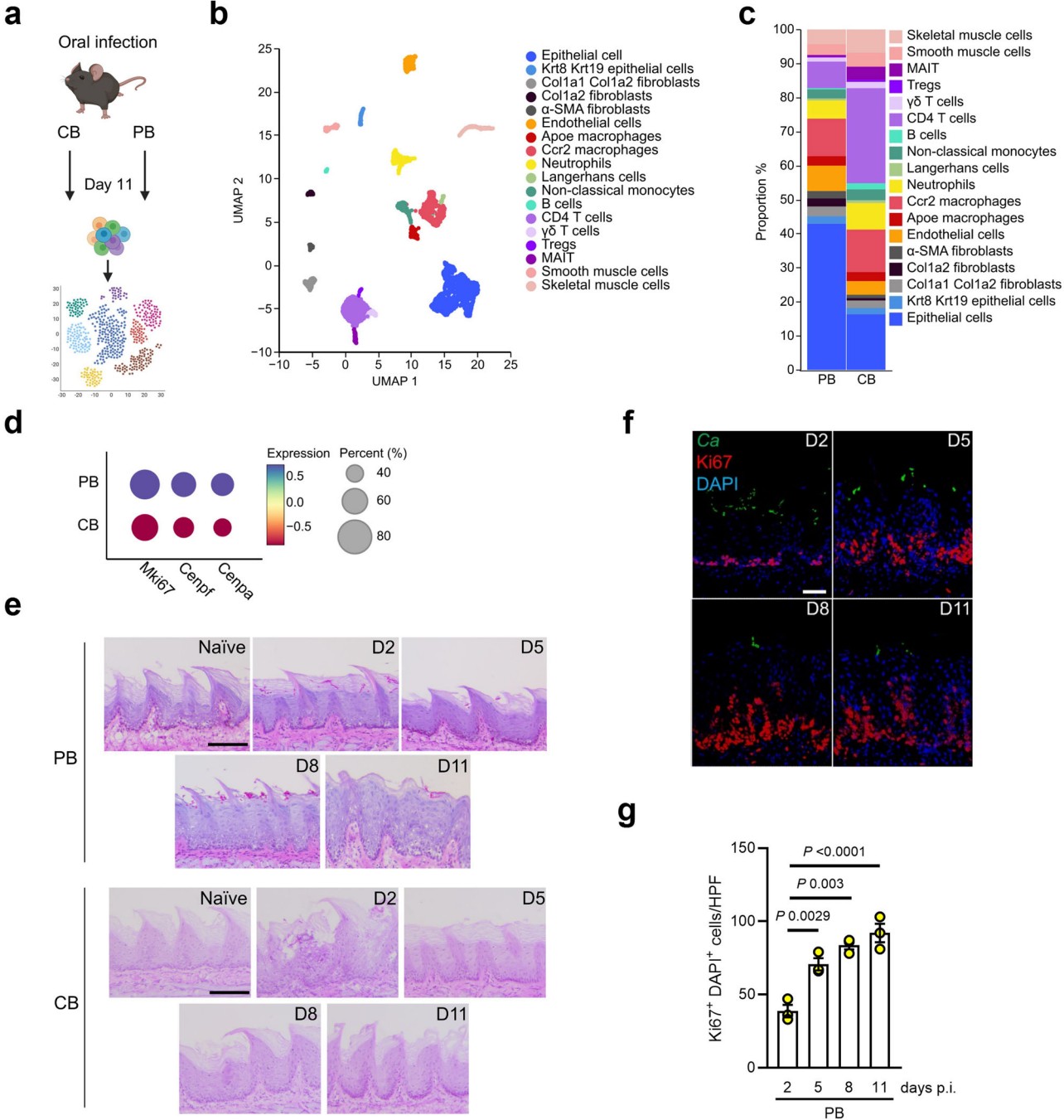

**Fig. 1 | Persistent oral *C. albicans* colonization induces epithelial remodeling.**
**a** Experimental setup for single cell collection and sequencing. Created in BioRender. https://BioRender.com/mxe3dcy. **b** Cell types identified in the oral mucosa after 11 days of PB and CB infection with UMAP projections of scRNA-seq data. **c** Proportions of identified cell types separated by CB and PB infection. **d** Expression of *Miki67*, *Cenpf*, and *Cenpa* in epithelial cells of CB- and PB-infected mice. **e** Representative pictures of PAS staining of CB- and PB-infected mice over time. Each experiment was repeated independently two times with similar results. Scale bar 100 µm. **f** Representative pictures of Ki67 staining of PB-infected mice over time. Scale bar 50 µm. **g** Quantification of Ki67+ DAPI+ cells PB-infected mice over time. *N* = 3; combined data of two independent experiments. For each animal (three mice per group), one section per tongue was randomly selected and stained. Data are presented as mean values +/− SEM. Ordinary one-way ANOVA. CB clearance-biased, PB persistence-biased.

cellular origin of IL-22. Our scRNA-sequencing dataset suggested that CD4+ T cells highly express *Il22* during persistent fungal colonization (Fig. 3e). Using *IL22TdTomato* reporter mice, we confirmed that CD4+ T cells are the major source of IL-22 during colonization with PB *Candida* (Fig. 3f). Next, we evaluated the localization of CD4 T cells during fungal persistence within the oral mucosa. CD4+ T cells were exclusively found in the epithelial and submucosal layers of PB-infected mice at the onset of colonization, which differed markedly

from CB- or Sham-infected mice (Fig. 3g, Supplementary Fig. S8) suggesting that persistent fungal stimulation is required to recruit and retain CD4 T cells in submucosal layers. Within the helper T cells, IL-22 is mainly produced by Th17 and Th22 subsets[39]. Ex vivo stimulation of mucosal resident CD4+ T cells from CB- and PB-infected mice showed that during fungal persistence Th17 and Th22 cells are the major source of IL-22, while Th22 cell frequencies increased (Fig. 3h–j).

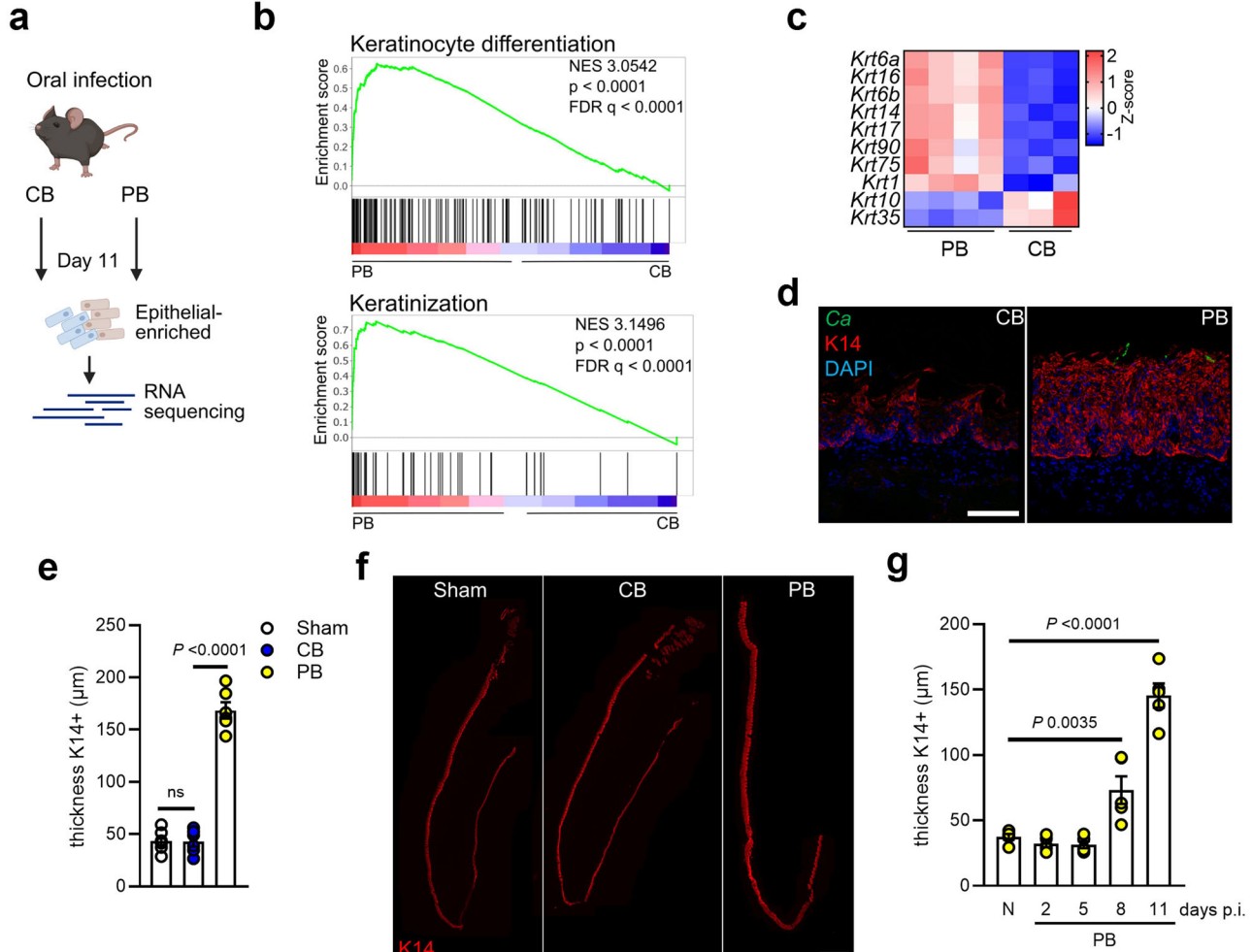

**Fig. 2 | Oral *Candida* persistence drives K14 epithelial subset expansion.**
**a** Experimental setup for sequencing of epithelial-enriched tissues. Created in BioRender. https://BioRender.com/7xtj9al. **b** GSEA of keratinocyte differentiation and keratinization sequencing data of epithelial-enriched tissues. $N = 3$–4. **c** Z-scores of keratin genes in epithelial-enriched tissues after CB and PB infection (Day 11). $N = 3$ (CB) and 4 (PB). **d** Representative pictures of K14 staining of CB- and PB-infected mice after 11 days. Scale bar 50 μm. **e** Quantification of K14 thickness in Sham-, CB, and PB-infected mice after 11 days. $N = 6$; combined data of two independent experiments. For each animal (six mice per group), one section per tongue was randomly selected and stained. Ordinary one-way ANOVA. Data are presented as mean values +/− SEM. **f** Representative pictures of K14 staining of whole tongues of Sham, CB- and PB-infected mice after 11 days. Scale bar 800 μm. **g** Quantification of K14 thickness in PB-infected mice over time. $N = 5$; combined data of two independent experiments. For each animal (five mice per group), one section per tongue was randomly selected and stained. Ordinary one-way ANOVA. Data are presented as mean values +/− SEM. CB clearance-biased, PB persistence-biased.

## Epithelial expansion depends on IL-22 signaling via non-canonical gp130 receptor complexes

The IL-17- and the IL-22 receptors, which mediate anti-*C. albicans* immunity during acute OPC, play distinct and restricted roles in distinct sublayers of the stratified oral epithelium[36]. During PB-*Candida* colonization, IL-22RA1 protein expression extended into the suprabasal layer (Supplementary Fig. S9) consistent with K14 expression (Fig. 2). IL-22 has been implicated in multiple aspects of epithelial barrier function and wound repair, including regulation of cell growth[40]. Accordingly, IL-22 induced proliferation of human oral epithelial cells in a dose-dependent manner, while high cytokine concentrations inhibited oral epithelial cell growth (Fig. 4a, Supplementary Fig. S10). Similarly, IL-22 deficient mice and IL-22 depletion during PB colonization reduced K14 epithelial layer expansion and proliferation (Fig. 4b–d, Supplementary Fig. S11), as well as resistance against fungal outgrowth (Fig. 4e) in the setting of intact IL-17 signaling (Supplementary Fig. S12). IL-22-mediated proliferation required JAK-TYK2-STAT3 signaling in human oral epithelial cells (Supplementary Fig. S13). Importantly, tonic STAT3 activity was required for proliferation in unstimulated oral epithelial cells (Supplementary Fig. S13). Classical IL-22 signal transduction is mediated by binding of the cytokine to a receptor complex consisting of IL-22RA1 and IL-10RB[27]. However, while mice deficient for either receptor, IL-22RA1 and IL-10RB respectively, were more susceptible to PB *Candida* outgrowth (Fig. 4f), the K14 layer surprisingly still underwent expansion in the absence of either receptor chain (Fig. 4g, Supplementary Fig. S14). Notably, IL-10 signals through a heterotetrameric complex comprising of IL10Rα and IL10RB but was dispensable for fungal control and epithelial remodeling (Supplementary Fig. S15). Recently, studies in synthetic cytokine biology have challenged the classical view of IL-22 signaling via a heterodimeric IL-22RA1 and IL-10RB complex. Synthetic cytokine receptor chains of IL-22RA1 and IL-10RB form functional heterodimeric receptor signaling complexes with an IL-6 receptor chain of gp130 to induce STAT3 signal transduction[41]. Proximity ligation assays and co-immunoprecipitation identified the classical heterodimeric IL-22RA1 and IL-10RB complex, while revealing that IL-10RB, as well as IL-22RA1, form receptor complexes with gp130 in human oral epithelial cells (Fig. 4h, Supplementary Fig. S16). We tested

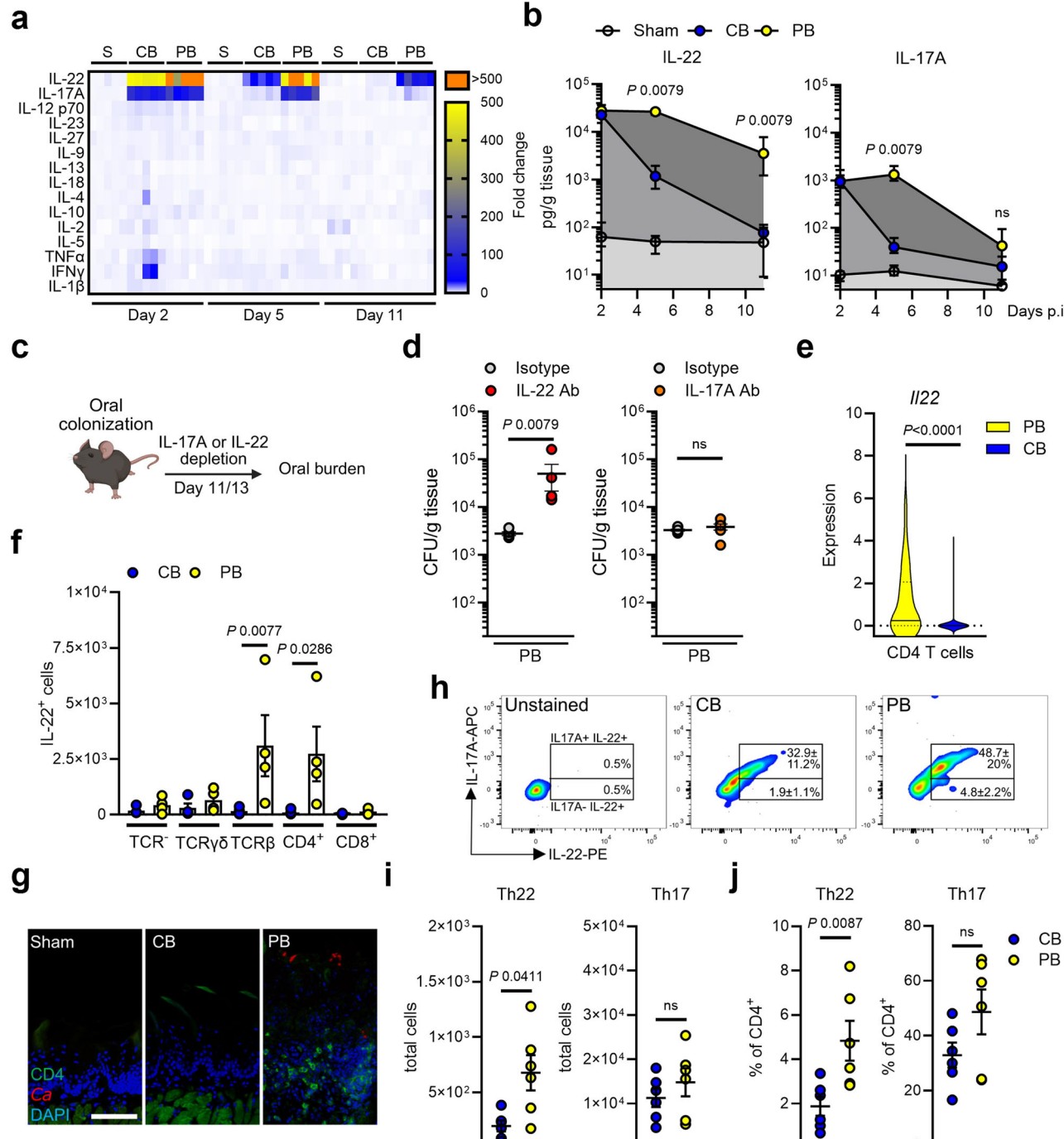

**Fig. 3 | Th17/Th22 cells produce IL-22 during *C. albicans* colonization. a** Heat map presented in fold change of various cytokines during CB, PB, and Sham infection over time *N* = 10; combined data of two independent experiments. Two-tailed Mann–Whitney Test. **b** Levels of IL-17A and IL-22 in CB-, PB-, and Sham-infected mice; *N* = 10. Two-tailed Mann–Whitney Test. Data are presented as mean values +/− SEM. **c** Scheme of IL-17 and IL-22 depletion during PB-colonization. Created in BioRender. https://BioRender.com/kf96h0q. **d** Oral fungal burden of mice colonized with PB after treatment on day 11 and 13 with anti-IL-17A or anti-IL-22 antibodies. *N* = 6; combined data of two independent experiments. Two-tailed Mann–Whitney Test. The y-axis is set at the limit of detection (20 CFU/g tissue). Data are presented as mean values +/− SEM. **e** Expression of *Il22* in CD4 T cells identified in the single cell RNA-sequencing data set. Wilcoxon matched-pairs signed rank test. *N* = 198 (PB) and 498 (CB) cells from sc-RNAseq data. Data are

presented as Violin plot including median and quartiles. **f** Quantification of IL-22 expressing cells after CB and PB infection (day 11) using *IL22TdTomato* reporter mice. *N* = 4, combined data of two independent experiments. Ordinary one-way ANOVA corrected for multiple comparison. Data are presented as mean values +/− SEM. **g** Representative immunofluorescence pictures of localization of CD4 T cells in tissues after 11 days of infection. Scale bar 50 μm. Each experiment was repeated independently three times with similar results. **h** IL-17A and IL-22 levels in CD4+ cells. Th17 cells were identified as CD4 + IL-17A + IL-22+. Th22 cells were identified as CD4 + IL-17A- IL-22+. **i** Total number and proportions **j** of Th17 and Th22 cells after 11 days of infection. *N* = 6; combined data of two independent experiments. Two-tailed Mann–Whitney Test. Data are presented as mean values +/− SEM. S Sham, CB clearance-biased, PB persistence-biased.

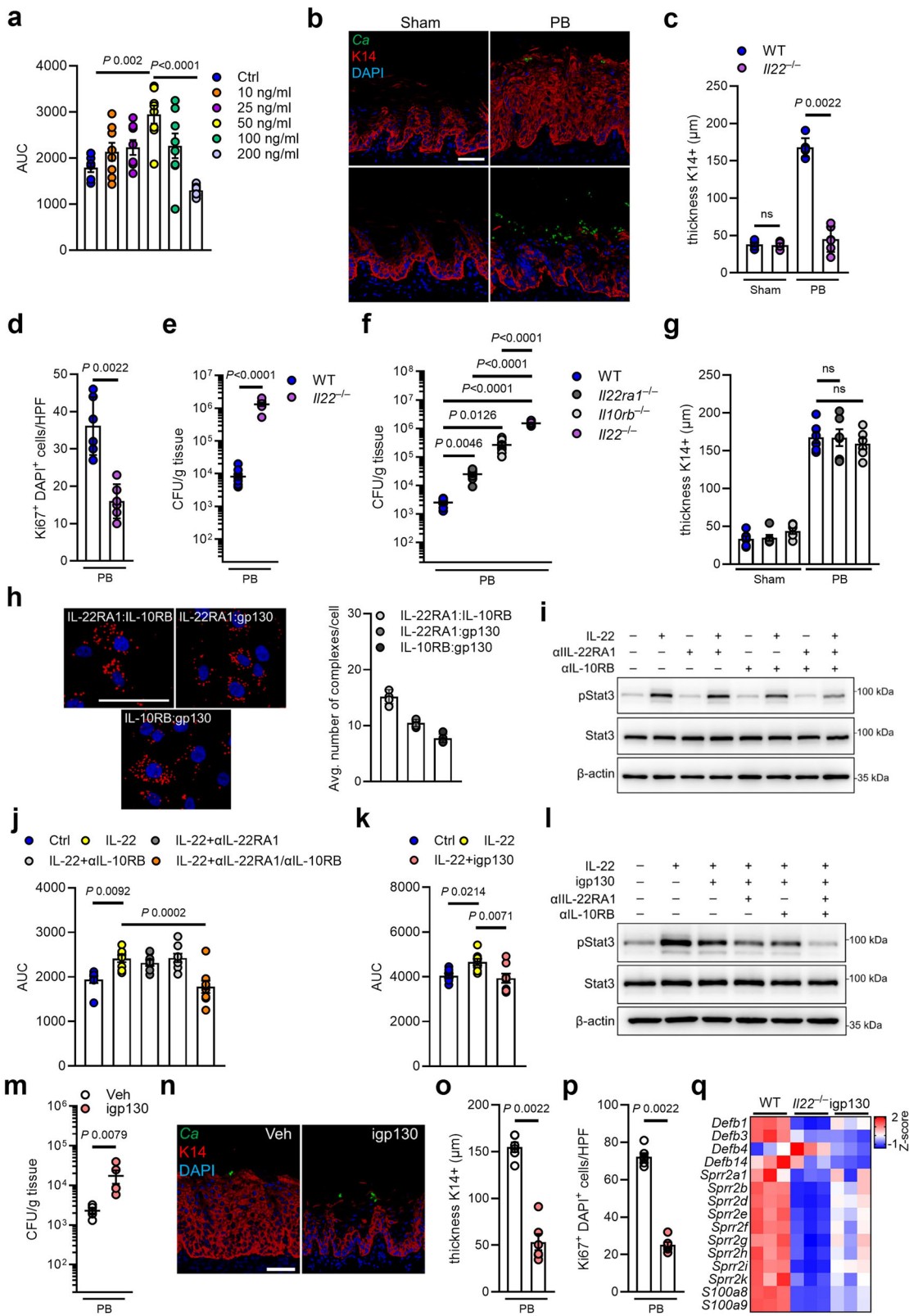

whether antibody blockade of either receptor chain, IL22RA1 and IL-10RB, would abolish STAT3 signal transduction in human oral epithelial cells. While antibodies blocking IL-22RA1 or IL-10RB induced similar STAT3 activation following IL-22 exposure, treatment with both antibodies reduced epithelial activation and IL-22 mediated proliferation (Fig. 4i, j, Supplementary Fig. S17). Next, we determined the contribution of gp130 signaling to IL-22-mediated oral epithelial

proliferation. Pharmacological inhibition of gp130 signaling reduced IL-22-mediated oral epithelial proliferation in vitro (Fig. 4K), while reducing STAT3 activation (Fig. 4l, Supplementary Fig. S17). IL-6 binds to membrane-bound IL-6R to induce signaling via gp130[42]. However, IL-22-mediated epithelial proliferation was independent of IL-6-gp130 signaling (Supplementary Fig. S18). Of note, STAT3 activity remained unchanged during inhibition of gp130 in intestinal epithelial cells

**Fig. 4 | Epithelial expansion requires non-classical IL-22 signaling. a** Area under the curve (AUC) of oral epithelial cells in response to different concentrations of IL-22. Growth was determined by confluence over time. $N = 8$; independent cultures. Ordinary one-way ANOVA. Data are presented as mean values +/− SEM. **b** Representative pictures of K14 staining of Sham- and PB-infected mice after 11 days. Scale bar 50 μm. **c** Quantification of K14 thickness in Sham- and PB-infected WT and $Il22^{-/-}$ mice at day 11. $N = 4$ (Sham) and 6 (PB); combined data of two independent experiments. Two-tailed Mann−Whitney Test. Data are presented as mean values +/− SEM. **d** Quantification of Ki67+ DAPI+ cells PB-infected mice over time. $N = 6$; combined data of two independent experiments. Two-tailed Mann−Whitney Test. Data are presented as mean values +/− SEM. **e** Oral fungal burden of WT and $Il22^{-/-}$ mice colonized with PB after 11 days. $N = 8$ ($Il22^{-/-}$) and 9 (WT); combined data of two independent experiments. Two-tailed Mann−Whitney Test. The y-axis is set at the limit of detection (20 CFU/g tissue). Data are presented as mean values +/− SEM. **f** Oral fungal burden of indicated mice colonized with PB after 11 days. $N = 7$; combined data of two independent experiments. Brown-Forsythe and Welch ANOVA. The y-axis is set at the limit of detection (20 CFU/g tissue). Data are presented as mean values +/− SEM. **g** Quantification of K14 thickness in indicated mice at day 11. $N = 6$; combined data of two independent experiments. Ordinary one-way ANOVA. Data are presented as mean values +/− SEM. **h** (Left) Representative Images for proximity ligation assay (PLA) for indicated receptor complexes. Scale bar 50 μm. Red dots indicate receptor complexes. (Right) Quantification of average number of complexes per cell. $N = 5$. **i** Representative immunoblot of STAT3 activation during IL-22 incubation in the presence of IL-22RA1, IL-10RB, or combination of IL-22RA1/IL-10RB blocking antibodies. Each experiment was repeated independently three times with similar results. **j** Area under the curve (AUC) of oral epithelial cells in response to IL-22 treatment in the presence of IL-22RA1, IL-10RB, or combination of IL-22RA1/IL-10RB blocking antibodies. Growth was determined by confluence over time. $N = 8$; independent cultures. Ordinary one-way ANOVA. Data are presented as mean values +/− SEM. **k** Area under the curve (AUC) of oral epithelial cells in response to IL-22 treatment in the presence of the gp130 inhibitor SC144 (igp130). Growth was determined by confluence over time. $N = 8$; independent cultures. Ordinary one-way ANOVA. **l** Representative immunoblot of STAT3 activation during IL-22 incubation in the presence of gp130 inhibitor, IL-22RA1, IL-10RB, or combination. Each experiment was repeated independently three times with similar results. **m** Oral fungal burden of mice treated with gp130 inhibitor (igp130) or vehicle control (Veh) colonized with PB after 11 days. $N = 6$; combined data of two independent experiments. Two-tailed Mann−Whitney Test. The y-axis is set at the limit of detection (20 CFU/g tissue). Data are presented as mean values +/− SEM. **n** Representative pictures of K14 staining of Veh- and igp130 treated mice after 11 days of PB infection. Scale bar 50 μm **o** Quantification of K14 thickness in Veh- and igp130 treated mice after 11 days of PB infection. $N = 6$; combined data of two independent experiments. Two-tailed Mann−Whitney Test. Data are presented as mean values +/− SEM. **p** Quantification of Ki67+ DAPI+ cells PB-infected mice in the presence and absence of gp130 signaling. $N = 6$; Two-tailed Mann−Whitney Test. Data are presented as mean values +/− SEM. **q** Z-scores of antimicrobial peptide genes in epithelial-enriched tissues in WT and $Il22^{-/-}$ mice, and WT mice treated with gp130 inhibitor (igp130) after PB infection on day 11. $N = 3$ mice per group. PB, persistence-biased.

---

(Supplementary Fig. S19), suggestive of organ-specific receptor complex signaling in response to IL-22. Inhibition of gp130 signaling during oral PB persistence increased the fungal burden and inhibited K14 expansion and remodeling (Fig. 4m−p). Thus, IL-22 signals via various receptor chain combinations to provide oral mucosal antifungal immunity during fungal immunosurveillance, while oral epithelial proliferation and expansion depend on non-canonical gp130 signaling receptor complexes. IL-22 cooperates with IL-17 to enhance AMP production, such as defensins, S100 proteins, and small proline-rich proteins (SPRRs) which synergistically protect mucosal surfaces from fungal invasion[12,43–45]. Epithelial-enriched RNA-seq revealed that mice lacking IL-22 or inhibition of gp130 signaling reduced expression of key antimicrobial peptides (Fig. 4q, Supplementary Fig. S20). Thus, oral epithelial expansion mediated by non-canonical IL-22 receptor signaling amplifies antifungal immunity by enhancing AMP expression.

## Transient epithelial expansion depends on development of *Candida*-specific immunity

While oral PB infection leads to epithelial expansion at the onset of colonization, at later time points of fungal persistence the K14 expansion is reset to levels comparable to the baseline cell turnover seen in naïve mice (Fig. 5a−c), suggesting that mucosal remodeling is transitory, regulated, and depends on an auxiliary mechanism. A unifying theme of susceptibility to mucocutaneous candidiasis is seen in both humans and mice with a variety of genetic defects within the IL-17 pathway[3,31,46]. The generation of organism-specific adaptive immune responses takes time to generate sufficient cells, due to the inherent demands for extensive proliferation and differentiation of naive cells into effector cells[47]. Thus, we tested if *Candida*-specific immunity is required to remodel the oral mucosal barrier in subsequent phases of colonization. During late stages of fungal persistence, increased numbers of *Candida*-specific IL-17 secreting cells in cervical lymph nodes (Fig. 5d) inversely correlated with reversion of the K14 barrier thickness (Fig. 5e) suggesting the requirement of *Candida*-specific immunity to reset barrier architecture. Importantly, $Rag1^{-/-}$ were more susceptible to fungal outgrowth at the onset and late stages of colonization (Fig. 5f), while K14 epithelial expansion remained high (Fig. 5g, h, Supplementary Fig. S21). Despite the absence of mature T cells, IL-22 expression is detected in $Rag1^{-/-}$ mice (Figure S21), indicating that non-mature T cell sources produce IL-22[48–51]. Of note, $Cd4^{-/-}$ mice

phenocopied the observed phenotypes of $Rag1^{-/-}$ during the onset of colonization (Supplementary Fig. S22). In the absence of mature T cells, ILC3s increased IL-22 expression at the onset of *Candida* colonization, likely contributing to epithelial expansion (Supplementary Fig. S21). We took advantage of the $Rag1^{-/-}$ model to test if *Candida*-specific T cell immunity is required to revers epithelial remodeling at later stages of colonization. We adoptively transferred purified CD4+ T cells from PB-colonized (PBCD4 T cells) and Sham-infected WT mice (SCD4 T cells) into PB-colonized $Rag1^{-/-}$ mice (Fig. 5i). PBCD4 T cells transferred into $Rag1^{-/-}$ mice reduced the fungal burden (Fig. 5j) and K14+ epithelial thickness (Fig. 5k, l) suggesting that *Candida*-primed CD4+ T cells are both required and sufficient of driving subsequent epithelial remodeling that occurs during fungal colonization. Collectively, we show that IL-22-mediated oral epithelial remodeling is transitory and requires *Candida*-specific immunity to trigger a successive mucosal remodeling event in later stages of colonization.

## Epithelial expansion is crucial to prevent fungal tissue invasion in neonates

Neonates exhibit differentially adapted immune responses when compared to adults[52], with important implications for immunity to OPC; indeed, oral thrush is commonly seen in infants[53], though the underlying basis for this is not well defined. In this regard, neonatal umbilical cord blood-derived CD4 T cells are intrinsically less able to differentiate into Th17 cells but rather tend to skew towards an IL-22 phenotype[54]. Here, we established a mouse model of neonatal fungal colonization to study early-life interactions between the oral mucosa and *Candida*, which occurs in humans (Fig. 6a, b). Consistent with our findings in adult animals, oral PB colonization in neonates induced K14 epithelial expansion (Fig. 6c, d) and CD4+ T cell infiltration at the onset of colonization (Supplementary Fig. S23). Neonatal mice deficient in IL-22 had increased oral fungal burden, abrogated K14 expansion, and decreased fungal-induced epithelial proliferation (Fig. 6f–h). Furthermore, IL-22 deficiency in neonates increased the depth of fungal tissue invasion (Fig. 6I, j) which was not observed in adult $Il22^{-/-}$ mice (Fig. 6j) suggesting that IL-22-mediated immunity and associated tissue remodeling plays a critical role preventing fungal invasion during early life, highlighting the importance of IL-22 in neonatal immune defense and epithelial protection.

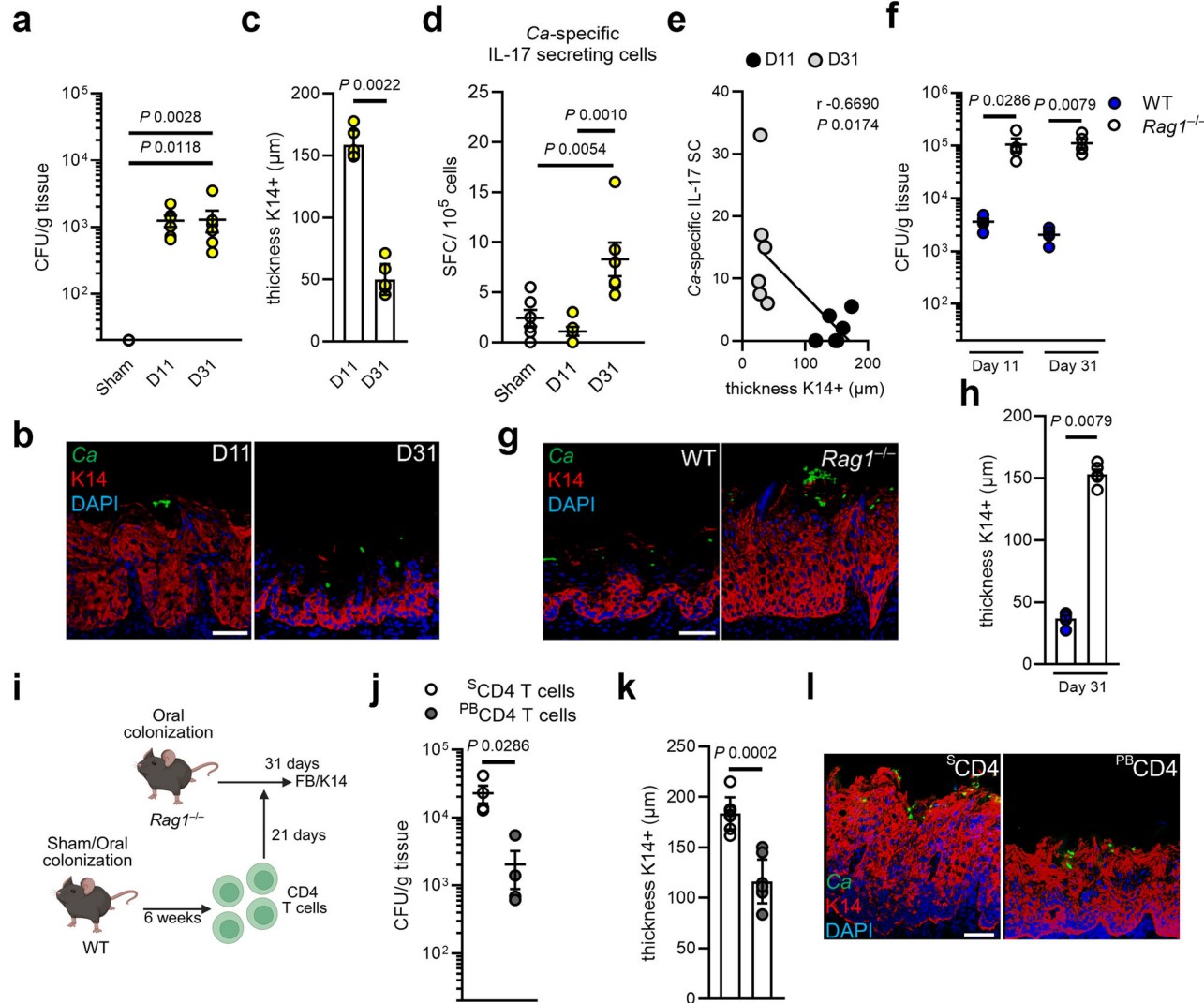

**Fig. 5 | Transitory epithelial K14 expansion depends on *Candida*-specific immunity. a** Oral fungal burden of mice colonized with PB after indicated time points. $N = 6$; combined data of two independent experiments. Two-tailed Mann−Whitney Test. The y-axis is set at the limit of detection (20 CFU/g tissue). Data are presented as mean values +/− SEM. **b** Representative pictures of K14 staining of PB-infected mice after 11 and 31 days. Scale bar 50μm. **c** Quantification of K14 thickness in PB-infected mice at indicated time points. $N = 6$; combined data of two independent experiments. Two-tailed Mann−Whitney Test. Data are presented as mean values +/− SEM. **d** Determination of IL-17 secreting cells isolated from cervical lymph nodes mice and stimulated with *Candida* antigen pools for 24 h using ELISpot. $N = 6$. Ordinary one-way ANOVA. Data are presented as mean values +/− SEM. **e** Correlation of K14 thickness and *Candida* (*Ca*)· specific IL-17 secreting cells. $N = 6$. Correlation was determined by Pearson. **f** Oral fungal burden of WT and *Rag1*[−/−] mice colonized with PB after 11 and 31 days. $N = 4$ (day 11) and 5 (day 31); combined data of three independent experiments. Two-tailed Mann−Whitney Test. The y-axis is set at the limit of detection (20 CFU/g tissue). Data are presented as mean values +/− SEM. **g** Representative pictures of

K14 staining of PB-infected mice after 31 days. Scale bar 50 μm. **h** Quantification of K14 thickness in PB-infected mice after 31 days. $N = 5$; combined data of two independent experiments. Two-tailed Mann−Whitney Test. **i** Scheme of CD4 T cell adoptive transfer in PB-colonized *Rag1*[−/−] mice. Created in BioRender. https://BioRender.com/lhit89u. **j** Oral fungal burden of *Rag1*[−/−] mice receiving CD4 T cells from Sham-infected ([S]CD4 T cells) WT mice and PB-colonized WT mice ([PB]CD4 T cells). $N = 4$; combined data of two independent experiments. Two-tailed Mann−Whitney Test. The y-axis is set at the limit of detection (20 CFU/g tissue). Data are presented as mean values +/− SEM. **k** Quantification of K14 thickness at day 31. $N = 4$ in duplicate; combined data of two independent experiments. Two-tailed Mann−Whitney Test. For each animal (four mice per group), two non-consecutive sections per tongue (sections from different parts of the tongue) were randomly selected and stained. Data are presented as mean values +/− SEM. **l** Representative pictures of K14 staining of *Rag1*[−/−] mice receiving [S]CD4 T cells and [PB]CD4 T cells. Scale bar 50 μm. Each experiment was repeated independently two times with two mice per group with similar results. PB persistence-biased.

## Discussion

Resistance to mucosal fungal infection is tightly linked to maintenance of barrier integrity[20,55]. The communication between epithelial and immune cells enables coordinated responses that maintain homeostasis and elicit host defenses[21,56]. Our findings support a paradigm by which fungal-induced immunosurveillance in the oral mucosa leads to transitory mucosal tissue remodeling and enhanced barrier resistance to fungal outgrowth.

The microbiome is a crucial factor for shaping and modulating immune system responses through cytokine signatures[57]. As such, our data unveil an IL-22-mediated pathway that critically regulates mucosal antifungal host defense in mice following continued exposure to fungi. We demonstrate that oral epithelial IL-22 signaling relies on various receptor chain combinations to mediate antifungal immunity, while gp130 receptor complexes drive epithelial remodeling and proliferation, which expands on reported tissue-specific protective roles of IL-

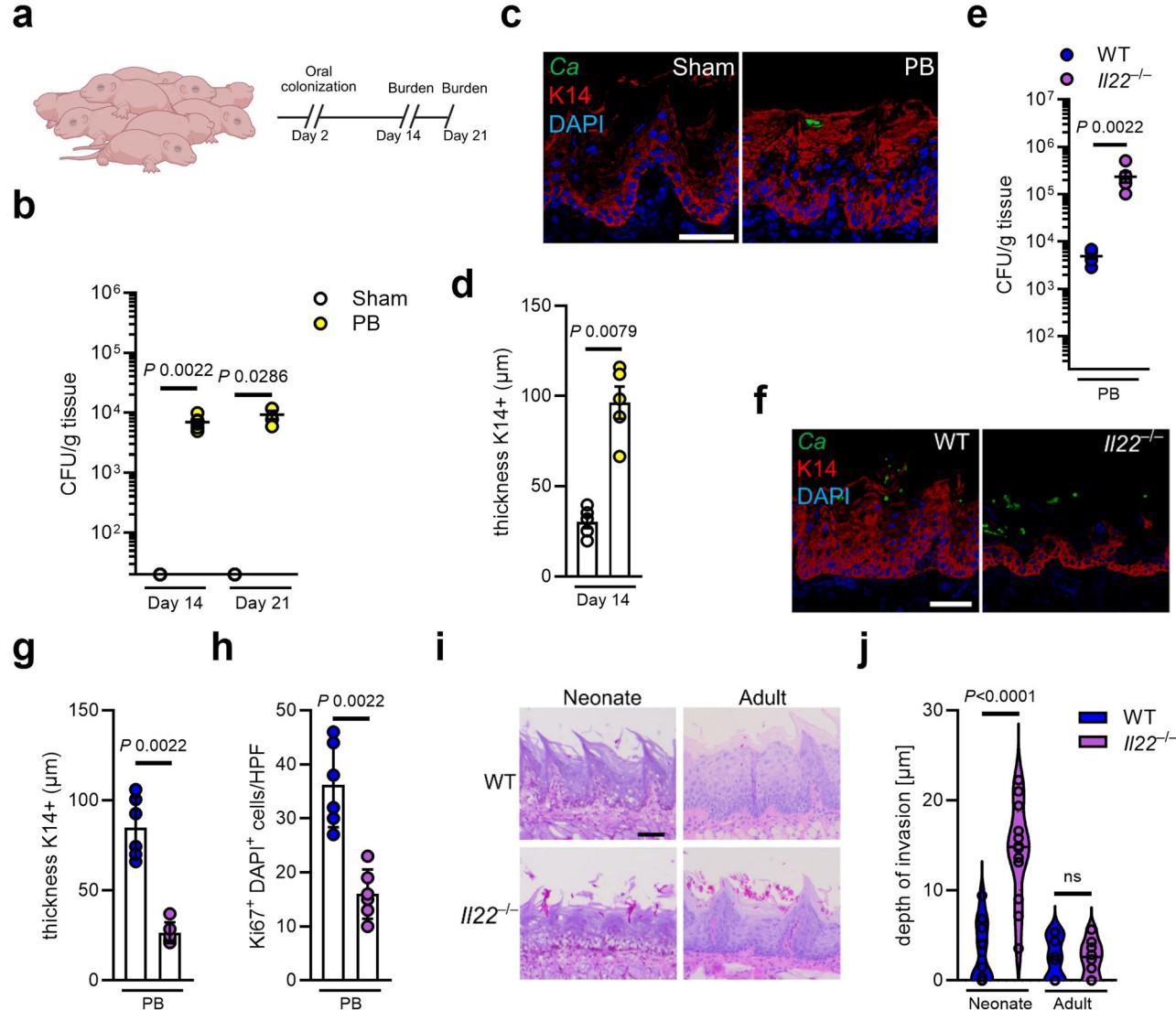

**Fig. 6 | Epithelial remodeling prevents fungal invasion during neonatal colonization. a** Experimental setup for neonatal colonization of wild type mice. Created in BioRender. https://BioRender.com/ga8mzi3. **b** Oral fungal burden of neonatal mice Sham-infected or colonized with PB after indicated time points. *N* = 4 (day 21) and 6 (day 14); combined data of two independent experiments. Two-tailed Mann−Whitney Test. The y-axis is set at the limit of detection (20 CFU/g tissue). Data are presented as mean values +/−SEM. **c** Representative pictures of K14 staining of Sham-infected or PB-infected mice at day 14 of life. Scale bar 50 µm. **d** Quantification of K14 thickness in Sham-infected or PB-infected mice at day 14 of life. *N* = 5; combined data of two independent experiments. Two-tailed Mann−Whitney Test. Data are presented as mean values +/− SEM. **e** Oral fungal burden of neonatal WT and *Il22*⁻/⁻ mice colonized with PB at day 14 of life. *N* = 6; combined data of two independent experiments. Two-tailed Mann−Whitney Test. The y-axis is set at the limit of detection (20 CFU/g tissue). Data are presented as mean values +/− SEM. **f** Representative pictures of K14 staining of WT and *Il22*⁻/⁻ mice colonized with PB at day 14 of life. Scale bar 50 µm. **g** Quantification of K14 thickness in neonatal WT and *Il22*⁻/⁻ mice colonized with PB at day 14 of life. *N* = 5; combined data of two independent experiments. Two-tailed Mann−Whitney Test. Data are presented as mean values +/− SEM. **h** Quantification of Ki67⁺ DAPI⁺ cells in neonatal WT and *Il22*⁻/⁻ mice colonized with PB at day 14 of life. *N* = 6; combined data of two independent experiments. Two-tailed Mann−Whitney Test. **i** Representative PAS staining of tongue tissues of neonatal (day 14 of life/day 12 of PB colonization) and adult (day 11 of PB colonization) WT and *Il22*⁻/⁻ mice. Scale bar 50 µm. **j** Quantification of depth of fungal invasion in neonatal WT and *Il22*⁻/⁻ mice colonized with PB at day 14 of life and adult mice after 11 days of PB colonization. *N* = 8 (adult) and 12 (neonate); combined data of four independent experiments. Two-tailed Mann−Whitney Test. Data are presented as Violin plots indicating median and quartiles. PB persistence-biased.

22 receptor signaling[58]. Our findings of a non-canonical IL-22 receptor complex that orchestrates epithelial remodeling define a mechanism of mucosal antifungal defense that amplifies antimicrobial peptide expression and expands the current understanding of IL-22 signaling. The IL-22 recognizing receptor complexes contribute to different degrees to oral antifungal immunity indicated by diverse level of fungal dysbiosis. These findings align with clinical phenotypes of inborn errors of immunity in IL-10RB, in which patients are resistant to oral candidiasis[59] while a small fraction of patients with gp130-dependent hyper-IgE syndrome develop chronic mucocutaneous candidiasis[60,61].

Future studies will be required to determine additive and compensatory mechanisms of the IL-22RA1, IL-10RB, and gp130 epithelial receptor complexes during fungal colonization and infection in the oral mucosa. Inhibition of gp130, as well as dual receptor blockage of IL-22RA1 and IL-10RB, impairs IL-22-mediated STAT3 phosphorylation and epithelial proliferation, while some transcription factor activity remains. T cell differentiation depends on optimal STAT3 phosphorylation[62], suggesting that a similarly fine-tuned mechanism may be required in epithelial cells to promote proliferation. On the other hand, IL-22 signaling via IL-22RA1, IL-10RB, and gp130 complexes

is organ- and context dependent since IL-22-mediated STAT3 activation in intestinal epithelial cells occurs independent of gp130.

*Candida*-responsive CD4[+] T cells, predominantly Th17, are primed in healthy individuals through commensal exposure[4,63,64]. Interestingly, our data show that Th22 cells emerge during fungal colonization despite the absence of classical polarizing cytokines. Dendritic cells can directly stimulate CD4[+] T cells to produce IL-22 through mechanisms independent of IL-6 and IL-23[65]. This suggests that alternative tissue-specific cues, possibly derived from persistent fungal antigen exposure, are sufficient to drive IL-22 production. These findings challenge the conventional paradigm of Th22 differentiation and highlight a non-inflammatory route of T cell activation that may be critical for maintaining mucosal homeostasis during commensal colonization. In naïve hosts, early anti-*Candida* immunity relies on epithelial sensing and innate responses, with Flt3L-dependent and CCR2-dependent dendritic cells coordinating antigen presentation and T cell priming[13,66,67]. Epithelial remodeling follows a two-phase pattern: an initial IL-22-driven expansion that enhances resistance, followed by reversal as *Candida*-specific IL-17 immunity establishes control (Supplementary Fig. S24). While CD4[+] T cells dominate IL-22 production in immunocompetent hosts, ILC3s compensate in the absence of T cells, sustaining epithelial proliferation in *Rag1*[−/−] mice. These findings underscore the critical role of adaptive immunity in terminating epithelial expansion and achieving long-term fungal control.

During early life, newborns encounter an abundance of antigenic challenges derived from commensal and pathogenic organisms. Although 5–7% of infants develop oral candidiasis[53], aberrant cellular innate immune responses and an inexperienced adaptive immune system do not elaborate high resistance against fungal outgrowth. Thus, neonates may exhibit a degree of immunological tolerance to fungal colonization to prevent harmful immune reactions in mucosal tissues. After birth, the neonatal epithelium is permeable and lacks pattern recognition receptors[68,69], suggesting that neonatal oral epithelial cells have reduced sensitivity to microbial exposure relative to the adult to prevent excessive immune reactivity[69]. However, the microbiota induces a temporary protective mechanism in the neonatal oral epithelium by increasing salivary antimicrobial components that shape the oral bacterial composition and burden[68,69]. Here we show that neonatal oral *Candida* colonization and consequently IL-22-mediated mucosal remodeling prevents fungal epithelial invasion suggesting that protective compensatory mechanisms present in adults that likely are underdeveloped or absent in early life. Given that neonates mount a vigorous T cell response to microbial exposure rather than developing immunological memory[70], oral mucosal remodeling may play a key role to control colonizing fungi during early life. Understanding such early mechanisms are important for human health, because a deficiency of this remodeling process could translate into oral and systemic pathologies in adult life. Future studies will be required to determine epithelial remodeling and turnover to age-related differences in bystander and antigen-specific T cell activation to maintain mucosal homeostasis.

Our study provides a mechanistic framework for IL-22–driven epithelial remodeling during oral *Candida* colonization, yet several areas merit future exploration. Murine models offer a tractable platform to define immune mechanisms, but they may not fully capture the complexity of human mucosal immunity and epithelial physiology. Likewise, our focus on the oral mucosa leaves open whether gp130-dependent noncanonical IL-22 signaling generalizes to other barrier sites. At the level of T cell biology, the molecular cues that promote IL-22 production by CD4[+] T cells in the relative absence of classical polarizing cytokines remain to be defined. In addition, while epithelial expansion is transient and reverses as *Candida*-specific immunity emerges, the potential cumulative effects of repeated remodeling on barrier integrity and cross-protection against other pathogens remain unexplored. Although we surveyed multiple clinical isolates, a deeper analysis of how fungal strain diversity and host genetic variation shape IL-22 signaling and barrier remodeling will strengthen translational relevance. Finally, while our neonatal colonization model highlights the importance of IL-22–mediated epithelial remodeling in early life, age-specific differences in immune responses and epithelial barrier properties may limit direct extrapolation to human infants. Future studies should investigate how developmental factors, such as distinct T cell polarization and epithelial turnover, influence IL-22 signaling during the neonatal period and whether these mechanisms adapt throughout postnatal development. Addressing these questions will refine the generalizability of our findings and inform strategies to harness IL-22 pathways for mucosal health and antifungal defense.

Collectively, our findings identify oral mucosal remodeling through non-canonical IL-22 receptor signaling as a critical determinant of resistance to mucosal fungal infection and highlight the importance of tissue-specific immune responses in the control of infectious disease.

## Methods

### Ethics statement

All animal work was approved by the Institutional Animal Care and Use Committee (IACUC) of the Lundquist Institute at Harbor-UCLA Medical Center and University of Pittsburgh.

### Organisms and cell lines

The *C. albicans* strains SC5314 (clearance-biased; CB), and CA101 (persistence-biased; PB) and 11 clinical isolates (oral isolates[71,72] TW1, 529L; catheter-associated isolates[73] SJCA1, SJCA3, SJCA5, SJCA7, SJCA9; vaginal isolates[73,74] JS1, JS3, JS7 and JS26) were used in the experiments. For in vivo experiments, strains were serially passaged three times in YPD broth. The OKF6/TERT-2 cells have been authenticated by RNA-sequencing[12], and have been tested for mycoplasma contamination. The intestinal epithelial cell line C2BBe1 was purchased from ATCC.

### Subject details

For in vivo animal studies, age-and sex matched mice were used. Animals were bred and housed under pathogen-free conditions at the Lundquist Institute in ventilated cages at $22 \pm 1\,°C$, $55 \pm 10\%$ relative humidity, 12 h/12 h dark/light cycle, with free access to food and water. Animals were randomly assigned to the different infection or treatment groups. Researchers were not blinded to the experimental groups because the endpoints (fungal burden, histology, cytokine levels) were objective measures of disease severity. *Il22ra1*[E2a-cre] (referred to as *Il22ra1*[−/−]) mice were provided by Jay K. Kolls[75]. *IL-22TdTomato*[76] mice were bred at the University of Pittsburgh. *Il10rb*[−/−] (B6.129S2-*Il10rb*[tm1Agt]/J)[77], *Il22*[iCre/iCre] (C57BL/6-*Il22*[tm1.1(icre)Stck]/J; referred to as *Il22*[−/−])[78], *Il10*[−/−] (B6.129P2-*Il10*[tm1Cgn]/J)[79], and *Rag1*[−/−] (B6.129S7-*Rag1*[tm1Mom]/J; non-leaky)[80] mice were purchased from the Jackson Laboratory, and maintained at the Lundquist Institute. C57BL/6 wild type mice were purchased from the Jackson Laboratory and cohoused with breeders for at least 1 week before the experiments.

### Mouse model of oropharyngeal candidiasis

For inoculation, immunocompetent animals (7–8-week-old male and female mice) were sedated, and a swab saturated with $2 \times 10^7$ *C. albicans* cells was placed sublingually for 75 min. For colony-forming unit (CFU) enumeration the tongues were harvested, weighed, homogenized and quantitatively cultured. Neonatal mice were generated using standard breeding procedures and 8-week-old adult mice. To colonize newborn mice, 2 days after birth neonates were given orally 10 μL of $2 \times 10^7$ *C. albicans* cells (CA101) in HBSS. Mice were euthanized on day 14 and 21 of life. For antibody depletion wild-type mice were treated intraperitoneally with 200 μg of anti-IL-22 (IL22JOP, Invitrogen) and 200 μg of anti-mouse IL-17A (17F3, BioXCell), or corresponding concentrations of isotype controls (Rat IgG2a kappa, eBR2a,

mouse IgG1 isotype, MOPC-21) on day 11 and 13 post oral PB-infection. For pharmacological gp130 inhibition, mice were treated daily starting one day prior PB colonization with 10 mg/kg SC144 (SelleckChem) in veterinary 0.9% Sodium Chloride (Vetivex). In some experiments, fecal pellets were collected, homogenized, and plated on SAB agar supplemented with chloramphenicol (80 µg/ml) and streptomycin (100 µg/ml). To determine dissemination after oral colonization, mice were given a subcutaneous injection of 25 mg/kg triamcinolone (Kenalog-10, Bristol-Myers Squibb Company) on day −1, +1, and +3 relative to infection. On day 4, livers were harvested, weighed, homogenized and quantitatively cultured.

## Single cell RNA-sequencing

Mice were orally infected with *C. albicans* as described above. After 11 days of infection, the animals were administered a sublethal anesthetic mix intraperitoneally. The thorax was opened, and a part of the rib cage was removed to gain access to the heart. The vena cava was transected and the blood was flushed from the vasculature by slowly injecting 10 mL PBS into the right ventricle. The tongue was harvested and cut into small pieces in 100 µL of ice-cold PBS. 1 mL digestion mix (4.8 mg/ml Collagenase IV; Worthington Biochem, and 200 µg/ml DNase I; Roche Diagnostics, in 1x PBS) was added after which the tissue was incubated at 37 °C for 45 min. The resulting tissue suspension was then passed through a 100 µm cell strainer. For one sample, single cells of two mice were combined. Single cell library preparation and sequencing were conducted by Singulomics (Bronx, USA). Cryopreserved viable single cell suspensions were thawed, washed, resuspended in cell culture media with 0.04% bovine serum albumin and counted. Viable cell suspensions were then loaded into the Chromium Controller (10x Genomics) to generate gel beads-in-emulsion (GEM), with each GEM containing a single cell as well as barcoded oligonucleotides. We targeted 10,000 cells to be captured per sample. Next, the GEMs were placed in the SimpliAmp 96-well Thermal Cycler (Thermo Fisher Scientific) and reverse transcription was performed in each GEM (GEM-RT). After the reaction, the complementary cDNA was amplified and cleaned using Silane DynaBeads (10x Genomics) and the SPRI select Reagent kit (Beckman Coulter). Amplified full-length cDNAs from poly-adenylated mRNA were then used to generate a 3′ Gene Expression library (Chromium Next GEM 3′ Single Cell Reagent kits v3.1, dual index) following the manufacturer's instructions (10x Genomics). Amplified cDNAs and the libraries were measured using Qubit dsDNA HS assay (Thermo Fisher Scientific) and quality was assessed using BioAnalyzer (Agilent Technologies). The libraries were sequenced with ~200 million PE150 reads per sample on Illumina NovaSeq 6000, and fastq files of two samples were generated using Illumina bcl2fastq. Fastq files were subsequently processed using 10x Genomics Cell Ranger analytical pipeline (v4.0.0) and mouse mm10 reference. Cellranger aggr was used to aggregate outputs from both the libraries and the aggregated filtered processed files were finally taken as input to do a quality control check and further downstream clustering analysis using Cellenics/Trailmaker software (Parse Biosciences). After quality control to remove low-quality cells expressing high mitochondrial gene signatures and exclude doublets, Louvain clustering identified 20 subgroups in our samples (Supplementary Fig. S2). Cluster 1 and 10 were excluded from further analysis due to enriched lncRNA Gm42418 which is associated with ribosomal RNA contamination. The high-throughput sequencing data from this study have been submitted to the NCBI Sequence Read Archive (SRA) under accession number PRJNA1176644.

## Epithelial-enriched RNA-sequencing

The tongue tissue was placed in a 60 mm petri dish, and epithelial cells were isolated by gentle mechanical isolation. The isolated cells were placed in lysis buffer reagent and homogenized in Lysing Matrix C tubes (MPBio) containing a ¼″ ceramic bead (MPBio). RNA extraction

was performed with the RiboPure Kit (Ambion), and RNA sequencing was performed by Novogene Corporation Inc. (Sacramento, USA)[81]. mRNA was purified from total RNA using poly-T oligo-attached magnetic beads. To generate the cDNA library the first cDNA strand was synthesized using random hexamer primer and M-MuLV Reverse Transcriptase (RNase H⁻). Second strand cDNA synthesis was subsequently performed using DNA Polymerase I and RNase H. Double-stranded cDNA was purified using AMPure XP beads and remaining overhangs of the purified double-stranded cDNA were converted into blunt ends via exonuclease/polymerase. After 3′ end adenylation a NEBNext Adaptor with hairpin loop structure was ligated to prepare for hybridization. In order to select cDNA fragments of 150-200 bp in length, the library fragments were purified with the AMPure XP system (Beckman Coulter, Beverly, USA). Finally, PCR amplification was performed and PCR products were purified using AMPure XP beads. The samples were read on an Illumina NovaSeq 6000 with ≥20 million read pair per sample. Downstream analysis was performed using a combination of programs including STAR, HTseq, and Cufflink. Alignments were parsed using Tophat and differential expressions were determined through DESeq2. KEGG enrichment was implemented by the ClusterProfiler. Gene fusion and difference of alternative splicing event were detected by Star-fusion and rMATS. The reference genome of *Mus musculus* (GRCm38/mm10) and gene model annotation files were downloaded from NCBI/UCSC/Ensembl. Indexes of the reference genome was built using STAR and paired-end clean reads were aligned to the reference genome using STAR (v2.5). HTSeq v0.6.1 was used to count the read numbers mapped of each gene. The FPKM of each gene was calculated based on the length of the gene and reads count mapped to it. FPKM, Reads Per Kilobase of exon model per Million mapped reads, considers the effect of sequencing depth and gene length for the reads count at the same time[82]. Differential expression analysis was performed using the DESeq2 R package (2_1.6.3). The resulting P-values were adjusted using the Benjamini and Hochberg's approach for controlling the False Discovery Rate (FDR). Genes with an adjusted *P*-value < 0.05 found by DESeq2 were assigned as differentially expressed (cutoff fold change 1.5,).To allow for log adjustment, genes with 0 FPKM are assigned a value of 0.001. Correlation were determined using the cor.test function in R with options set alternative = "greater" and method = "Spearman". To identify the correlation between the differences, we clustered different samples using expression level FPKM to see the correlation using hierarchical clustering distance method with the function of heatmap, SOM (Self-organization mapping) and kmeans using silhouette coefficient to adapt the optimal classification with default parameter in R. We used clusterProfiler R package to test the statistical enrichment of differential expression genes in KEGG pathways. The high-throughput sequencing data from this study have been submitted to the NCBI Sequence Read Archive (SRA) under accession number PRJNA1176644.

## Immunofluorescence

To determine host cell localization in vivo, 15−30-µm-thick sections of OCT-embedded tongues were fixed with cold acetone. Next, the cryosections were rehydrated in PBS and then blocked using BSA. To detect K14 (#Ab181595, Abcam), K13 (#BSM-52053R, Bioss), Ki67 (#9126S, Cell Signaling), or CD4 (#100405, Biolegend) positive cells, the sections were incubated with a primary antibody overnight followed by 1 h incubation at room temperature with a secondary antibody either Alexa Fluor 488 or 568 goat anti- rabbit IgG (A11034, #A11011). To detect *C. albicans*, the sections were also stained with an anti-*Candida* antiserum (Biodesign International) conjugated with Alexa Fluor 568 (Thermo Fisher Scientific) for 1 h. To visualize the nuclei, the cells were also stained with DAPI (4′,6-diamidino-2-phenylindole). The sections (z-stack) were imaged by confocal microscopy. To enable comparisons of fluorescence intensities among slides, the same image acquisition settings were used for each experiment.

## Histology

Half-tongues were embedded in OCT (Fisher HealthCare, Houston, USA), frozen on dry ice, and stored at −80 °C. Sagittal cryosections (5 µm) were air-dried at room temperature, stained with periodic acid-Schiff (Sigma-Aldrich), and counterstained with hematoxylin (Sigma-Aldrich). Sections were mounted with a toluene solution (Permount®; Fisher Chemical). Images were analyzed with a phase-contrast microscope (Zeiss Axiostar) and Gryphax Software. To quantify fungal invasion, PAS-stained sections were randomly selected and imaged by light microscopy, and the depth growth of individual fungi relative to surface of the tongue were determined using PROGRES GRYPHAX® software version 1.1.8.153 (Jenoptik)[83,84].

## Cytokine and chemokine measurements in vivo

To determine the whole tongue cytokine and chemokine protein concentrations, mice were infected as described above. The mice were euthanized at various time points, and their tongues were harvested, weighed, and homogenized. The homogenates were cleared by centrifugation and the concentration of inflammatory mediators was measured using the Luminex multiplex bead assay (Invitrogen). In some experiments, IL-17A and IL-22 (R&D Systems) were determined by ELISA according to the manufacturer's instructions.

## Immunophenotyping

Mice were orally infected with *C. albicans* as described above. After different time points, the animals were administered a sublethal anesthetic mix intraperitoneally. The thorax was opened, and a part of the rib cage removed to gain access to the heart. The vena cava was transected and the blood was flushed from the vasculature by slowly injecting 10 ml PBS into the right ventricle. The tongue was harvested and cut into small pieces in 100 µl of ice-cold PBS. 1 ml digestion mix (4.8 mg/ml Collagenase IV; Worthington Biochem, and 200 µg/ml DNase I; Roche Diagnostics, in 1x PBS) was added after which the tissue was incubated at 37 °C for 30 min. The resulting tissue suspension was then passed through a 100 µm cell strainer. Cell suspensions were separated by Percoll gradient centrifugation as described before[15,36]. To determine IL-17A and IL-22, cell suspensions were stimulated with Pharmingen™ Leukocyte Activation Cocktail (BD Biosciences) for 5 h. For Th17 and Th22 cells, cells were washed and stained with CD4 antibody (BioLegend), for ILC3 cells, cells were washed and stained with FITC conjugated Gr-1, CD5, CD11b, CD45R (to eliminate unwanted cell populations), and CD3, CD90, Sca-1, CD127, CD95 (BioLegend) antibodies. For intracellular staining, cells were fixed with Cytofix/Cytoperm (BD Biosciences) and stained for 1 h with IL-17A (BioLegend) and IL-22 (BioLegend) antibodies. The stained cells were analyzed on FACSymphony A5 system (BD Biosciences), and the data were analyzed using FACS Diva and FlowJo software. Th17 cells were identified as singlets CD4+ IL-17A+ IL22+, Th22 cells were identified as singlets CD4+ IL-17A− IL22+ and ILC3 cells were identified as singlets Gr-1- CD5- CD11b- CD45R- CD3- CD90+ CD127+ Sca1+ CD95+ (Supplementary Fig. S25).

## ELISpot

Cervical lymph nodes (cLN) were isolated from PB-colonized or Sham-infected animals. cLN were digested with 2.4 mg/ml Collagenase I (Thermo Fisher) and 200 µg/ml DNase I (Roche Diagnostics) in 1× PBS. The resulting tissue suspension was then passed through a 70 µm cell strainer, washed, and live cells were determined. $1 \times 10^5$ cLN cells were plated on pre-coated PVDF plate, left unstimulated or were stimulated with 15 µg *Candida* peptide pool for 24 h at 37 °C and 5% $CO_2$. ELISpot was performed according to the manufacturer's instructions (ImmunoSpot). Plates were analyzed with a ImmunoSpot® Analyzer (CTL). For *Candida*- specific IL-17 detection, spots of *Candida* pool wells were determined and subtracted from spots determined in corresponding unstimulated wells.

The peptide pool (lysate) was generated from stationary cultures of PB strain CA101 as described elsewhere[85].

## Isolation of T cells and adoptive transfer experiments

CD4 T cells were harvested from WT mice (PB-colonized or Sham-infected 6 weeks after infection) for adoptive transfer into recipient *Rag1*−/− mice. The spleens and submandibular lymph nodes were harvested, enzymatically digested with PBS containing Collagenase IV (100 U/mL), DNase I (20 U/mL), 1% FBS, and 2.4 mg/ml Collagenase I (240 U/mL) and DNase I (400 U/mL) for 30 min. Single cell suspensions were passed through a 70-µm filter. After washing, negative CD4 T cell selection was performed using MojoSort™ Mouse CD4 T Cell Isolation Kit (#480006; BioLegend). Post-enrichment purity of CD4 T cells was >97.1%. $1 \times 10^6$ CD4 T cells were injected intravenously into *Rag1*−/− recipient mice 21 day after PB-colonization. After 10 days (day 31 post PB-colonization), CFUs and K14 thickness were determined.

## Live-cell imaging for cell proliferation

$1.2 \times 10^4$ OKF6/TERT-2 cells were plated in 96-well plates and incubated overnight at 37 °C, allowing them to settle down. Cells were treated without or with 10, 25, 50, 100, and 200 ng/ml of human IL-22 (PeproTech) in keratinocyte serum-free medium (KSF), and immediately placed in the IncuCyte SX5 Live Cell Analysis System (Sartorius, Göttingen, Germany). To identify the impacts of IL-22-mediated receptor complex and downstream signaling pathways, cells were treated with 5 µg/ml anti-hIL-22Rα1 (AF2770; R&D Systems), 5 µg/ml anti-IL-10RB (90220; R&D Systems) or in a combination, 1 µM TYK2 inhibitor (BMS-911543, SelleckChem), 0.5 µM (C188-9; SelleckChem), 5 µM STAT3 inhibitor (S3I-201; SelleckChem), 50 nM gp130 inhibitor (SC144; SelleckChem), and 5 µg/ml tocilizumab (A2012; SelleckChem). Cell proliferation was observed and automatically analyzed from acquired three images per well taken every four hours for a consecutive 72 h. Data was analyzed with the Incucyte live cell analysis software.

## Immunoblotting

OKF6/TERT-2 cells in 24-well tissue culture plates were switched to KSF medium without supplements for 1 h and treated with 50 ng/mL recombinant human IL-22 (PeproTech). Next, the cells were rinsed with cold HBSS containing protease and phosphatase inhibitors and detached from the plate with a cell scraper. After the cells were collected by centrifugation, they were boiled in sample buffer. The lysates were separated by SDS-PAGE, and phosphorylation and total proteins were detected by immunoblotting with specific antibodies against pSTAT3 (Tyr705, D3A7, #9145, Cell Signaling), STAT3 (D3Z2G, #12640, Cell Signaling), and β-actin (8H10D10, #3700, Cell Signaling). Each experiment was performed at least 3 times. The immunoblots were developed using enhanced chemiluminescence and imaged with a C400 digital imager (Azure Biosystems, Dublin, CA, USA). Uncropped immunoblots are presented in Supplementary Figs. S26, 27.

## Co-immunoprecipitation

OKF6/TERT-2 cells were grown in KSF medium with supplements to 70% confluence and cells were treated without or with 50 ng/ml human IL-22 (PeproTech). After 3 h. cells were incubated with fresh media containing 0.1 mM DSP-crosslinking (dithiobis(succinimidyl propionate) for 30 min. Cells were washed twice with PBS and re-incubated with fresh media containing 25 mM Tris-HCl for 15 min. Cells were lysed with ice-cold IP lysis buffer with protease/phosphatase inhibitors and protein concentrations were measured by Pierce BCA protein assay (Thermo Scientific). Co-immunoprecipitation was performed using the Pierce Classic Magnetic IP/Co-IP Kit according to the manufacturer's protocol (Thermo Scientific). In brief, 400 µg protein samples were added with 15 µg IL22RA1 (AF2770; R&D systems), IL10RB (MAB874; R&D systems), or gp130 antibody (362002; BioLegend). The

protein-antibody mixture was incubated overnight on a rotator at 4 °C to facilitate immune-complex formation. Following overnight incubation, 25 μL pre-washed Pierce Protein A/G magnetic beads were added to the protein-antibody mixture to capture the immuno-complex. The mixture was incubated for 1 h on a rotator at room temperature. Proteins were collected using a magnetic stand and separated by SDS-PAGE.

## Proximity ligation assay

$2.5 \times 10^5$ OKF6/TERT-2 cells were seeded onto fibronectin-treated coverslips in a 24-well plate overnight. The next day, cells were fixed with 2% paraformaldehyde for 10 min at room temperature. After washing and blocking, the cells were incubated with a goat anti-IL22RA antibody (#AF2770, R&D Systems, 5 μg/ml), a mouse anti-IL10RB antibody (#MAB874, Clone 90220, R&D Systems, 5 μg/ml), a rabbit anti-gp130 antibody (#37325, Cell Signaling, 5 μg/ml), or two of these antibodies in combination overnight at 4 °C. The interactions of IL22RA with IL10RB and the interactions of gp130 with either IL22RA or IL10RB were detected using the Duolink In Situ Red Starter Kit (DUO92101-1kit, Sigma-Aldrich) following manufacturer's instructions. The cells were imaged using a Leica TCS SP8 confocal microscope. Z-stacked images were obtained and overlaid using the Leica image processing software. To calculate the average number of complexes per cell, images were analyzed using ImageJ[86].

## Quantification and statistical analysis

At least three biological replicates were performed for all in vitro experiments unless otherwise indicated. Data were compared by Ordinary one-way ANOVA or Mann–Whitney corrected for multiple comparisons were appropriate using GraphPad Prism (V. 10.3) software. $P$ values < 0.05 were considered statistically significant.

## Reporting summary

Further information on research design is available in the Nature Portfolio Reporting Summary linked to this article.

# Data availability

The authors declare that the data supporting the findings of this study are available within the paper and the accompanying supplementary information files. Source data are provided with this paper. The high-throughput sequencing data from this study have been deposited with links to BioProject accession number PRJNA1176644 in the NCBI BioProject database https://www.ncbi.nlm.nih.gov/bioproject/PRJNA1176644. Source data are provided with this paper.

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

## Acknowledgements

We thank Jay K. Kolls (Tulane University) for providing the *Il22ra1$^{E2a-cre}$* mice, Salomé LeibundGut-Landmann (University of Zurich) for the *Candida albicans* commensal-like strain CA101, James G. Rheinwald (Dana-Farber/Harvard Cancer Center) for providing the OKF6/TERT-2 cell line, and members of the Division of Infectious Diseases at Harbor-UCLA Medical Center for critical suggestions. NIH grant R01DE031382, R21AI187999 (M.S.), R01AI177254 (S.G.F.), U19AI172713 (M.S., S.G.F.), R21AI159221, R56AI175328 (N.J.), F32AI186291 (M.E.C.), R01AI134796 (B.M.P.), R37DE022550 (S.L.G.), Division of Intramural Research of the NIAID (M.S.L.), California Institute for Regenerative Medicine Stem Cell Biology Training Grant EDUC4-12837 (N.M.).

## Author contributions

Conceptualization: M.S. Methodology: N.M., J.S., N.S., A.M., F.A., A.W., M.E.C., A.D., B.M.P., M.S.L., S.L.G., N.J., S.G.F., M.S. Investigation: N.M., J.S., N.S., J.M., A.M., F.A., A.W., M.S. Visualization: N.M., J.S., F.A., A.W., M.S. Funding acquisition: N.M., M.E.C., N.J., B.M.P., S.L.G., S.G.F., M.S. Project administration: M.S. Supervision: M.S. Writing – original draft: M.S. Writing – review & editing: N.M., J.S., N.S., J.M., A.M., F.A., A.W., M.S.L., B.M.P., S.L.G., N.J., S.G.F., M.S.

## Competing interests

The authors declare no competing interests.
