## [Transparent Peer Review file · Nature Communications]

Non-canonical IL-22 receptor signaling remodels the oral mucosal barrier during *Candida albicans* immunosurveillance

Corresponding Author: Dr Marc Swidergall

Version 0:

Reviewer comments:

Reviewer #1

(Remarks to the Author)

The manuscript by Millet et al. describes the immune response of mice infected orally with two different strains of *Candida albicans*.

The immune response and the activation of the epithelial barrier were investigated. The main highlighted response is the epithelial remodeling mediated by K14 expansion and the IL-22 production, particularly with the 101 *Candida albicans* strain.

The manuscript is well written, and the topic is clinically relevant, but I found essential conceptual and technical limitations that needed to be carefully clarified.

1. The main limitation of the manuscript is related to the definition of the two strains as 'commensal' and 'pathogenic'. Conceptually and scientifically, it is profoundly misleading and it needs to be explained or demonstrated in a completely different way. The oral model used by the authors is a simple mouse model of an acute infection, it is wrong to consider it as a 'commensalism' model. The infection is persistent with the 101 strain and this is not identified as a 'symbiotic' relation with the host but conversely as an opportunistic behaviour.

2. Figure S1 A is very surprising. Authors claim that the CL strain is not inducing inflammation but a non-inflammatory pathogen can't be associated with such a high persistent mouth colonization. Data should be presented by showing the standard deviation and if there is also dissemination in other tissues.

3. Authors have to clarify or modify the model, by considering also that *Candida albicans* is a human gut commensal and not properly a mouse commensal.

4. It is not clear why in figure 1E it is only shown the histology of mice infected with the CL strain and not the PL strain. Same issue for figure 1F, 1G, 1M.

5. In figure 1C it is surprising that single cell is not focused on Th17 cells but only Tregs are specifically stained, although Tregs were not the focus of the study.

6. Statistics is lacking in figure 1C

7. In figure 2, authors claim that CL is not inducing inflammation. This sentence is profoundly wrong, considering the pathogenic colonization overtime that the CL strain is inducing and also the evidence that the CL strain is inducing IL-17 and IL-22, two pro-inflammatory cytokines. Data coming from the assay in figure 2A should be repeated eventually by using conventional ELISA, since it is extremely surprising that IL-17 is induced and there are not T cell polarizing cytokines. In other words, do the author know how CD4+IL-22+ T cells were polarized if no inflammation was induced?

8. Figure 2 E is not showing statistics.

9. Figure F is showing a wrong statistic test.
10. Figure H is showing plots not clear. Gating strategy is not presented. CD4+IL-22+T cells should be shown.
11. Figure 4H is not clear. Why keratin is still there with no T cells?
12. Conclusions are in apparent contradiction with the main thesis. If you consider 101 a 'commensal' strain, how do you consider essential for survival the induction of IL-17 and IL-22 and keratinization? This is only suggesting an host essential (and already shown) response against a fungal pathogen.

Reviewer #2

(Remarks to the Author)

The manuscript by Millet et al. demonstrates that non-canonical IL-22 receptor complex (IL-22RA1/IL-10Rb/gp130) mediates commensal *Candida*-induced remodeling of the oral epithelium, facilitating immunosurveillance of the oral mucosa. Through comparison of pathogenic and commensal strains of *Candida*, the authors demonstrate that IL-22 is persistently upregulated until day 11, a process critical for inducing K14+ epithelial thickening via the IL-22RA1/IL-10Rb/gp130 complex, as supported by complementary knockout and inhibition experiments. This study provides valuable insights into non-canonical IL-22R signaling in antifungal responses, but certain aspects require further clarification to strengthen the conclusions. Below are my comments and suggestions.

1. Figures 2A and 2B indicate that IL-22 expression persists in the CL group but not in the PL group, with significantly higher levels at day 11 compared to IL-17A. However, Figure 2I shows that Th17 cells dominate the CD4+ T cell population, with an elevated frequency in the CL group, except for a single low outlier, which likely affects statistical significance. Meanwhile, Th22 cells are relatively low in frequency despite being significantly upregulated in the CL group. To better assess Th22 contributions, I recommend quantifying absolute cell numbers in addition to percentages of CD4+ T cells. Another possible explanation is that the Th17 population in Figure 2H might include IL-22/IL-17A double positive cells, which the authors should clarify or emphasize further.
2. Schönherr et al. (*Mucosal Immunology*, 2017) previously reported that IL-17 is critical for responses to both pathogenic and commensal *Candida* strains in oral infection models, with fungal burden increasing in both Il17ra and Il17rc knockout mice for both SC5314 and CA101 strains. This finding appears to conflict with the results shown in Figure 2D, where no fungal load differences were observed after IL-17 neutralization. The authors should discuss or speculate on why these observations differ.
3. Based on the fungal CFU and K14+ epithelial thickness data in Figure 3, thickening appears to be mediated primarily by non-canonical IL-22 receptor signaling through gp130. However, gp130 inhibition only partially impairs antifungal activity (Figure 3M), as CFU levels with gp130 inhibition are similar to those in Il22ra1 KO (Figure 3E), while K14+ thickening persists in the latter. This raises questions about the relationship between K14+ thickening and antifungal capacity. Could the authors elaborate on how these two phenomena are interconnected in the CL group?
4. The inverse correlation between *Candida*-specific IL-17+ cells and K14+ thickness in wild-type mice at days 11 and 31 is intriguing. The authors hypothesize that epithelial resetting is mediated by Th17 expansion, controlling CL *Candida*, while thickening persists in Rag1^{-/-} mice. Since CFU levels in wild-type mice are comparable between days 11 and 31, have the authors investigated CFU levels in Rag1^{-/-} mice at day 11? Additionally, given the IL22TdTomato reporter data indicating CD4+ T cells as the primary IL-22 producers during CL colonization (Fig. 2F), what factors might drive epithelial thickening in the absence of T cells in Rag1^{-/-} mice which lack IL-22 producing T cells?
5. The methodology for quantifying invasion depth in Figure 5I&J should be clearly detailed in the Materials and Methods section. Furthermore, the mechanisms underlying the increased fungal invasion observed in IL-22 KO neonates compared to adults remain unclear. Are fungal burdens different between neonates and adults? Have the authors examined the roles of other immune cells in colonization across these age groups? The findings in Figure 5 are primarily descriptive, and more mechanistic insights would strengthen this section.

Minor comments:

1. As all molecular analyses were performed on day 11, when *Candida* is absent in the PL group, how do the authors differentiate the effects of *Candida* presence on epithelial remodeling from baseline mucosal epithelial properties? For example, the lack of CD4+ T cells in the submucosal layer of the PL group (Fig. 2G) might simply reflect the absence of *Candida*. Clarification on this point would be helpful.
2. The statement "in CL-colonized mice, mucosal IL-22 levels remained high, while IL-17A levels were low at the onset of colonization" (Lines 114–115) should be revised, as IL-17A levels fluctuate significantly between days 2 and 5.
3. The legend for Figure S8 refers to treatment with both isotype and anti-IL-22 antibodies. However, the figure only includes PL and CL groups without showing data for isotype controls. The authors should ensure all experimental controls are represented.
4. Immunofluorescence images of the control group in Figure S6 would be informative. The aggregated IL-22RA1 signal in the PL group appears less pronounced in the CL group. Is this difference due to prior *Candida* infection, or is it also observed in uninfected controls?
5. The apparent toxicity or growth inhibition caused by the STAT3 inhibitor (Figure S10C) should be addressed by including a control group treated with the inhibitor alone. This would rule out potential off-target effects or the possibility that STAT3 signaling is essential for epithelial growth or survival.
6. The function of tocilizumab as an IL-6R-neutralizing antibody should be stated in the text or the legend for Figure S15.
7. The specificity of gp130 signaling in oral epithelium is an interesting observation. Have the authors examined whether similar mechanisms are involved in other mucosal tissues, such as the vaginal epithelium, another major colonization site for *Candida*?
8. It would be interesting to determine how long CL *Candida* can colonize the oral mucosa. In Figure 4A, fungal burden

remains comparable between days 11 and 31, while IL-17+ cell numbers significantly increase by day 31, suggesting immune resetting as proposed by the authors. How do the authors integrate this inverse relationship between IL-17+ cells and K14+ thickening in the context of long-term colonization?

Reviewer #3

(Remarks to the Author)

In this manuscript, Millet et al. report that oral colonization with a fungal component of the human mouth mycobiota elicits remarkable changes in the structure of the oral epithelium in mice. Specifically, the authors show that the tongue's epithelial layer expands in a process dependent on IL-22 signaling upon colonization with the oral commensal *Candida albicans* isolate 101, a strain that can persist for weeks/months in the murine tongues. Furthermore, Millet et al. demonstrate that IL-22 signaling depends on a non-canonical receptor complex composed of gp130 coupled to IL-22RA1 and IL-10RB. Finally, using a mouse model of neonatal fungal colonization, the authors propose that IL-22-mediated immunity and tissue remodeling have key roles controlling fungal invasion early in life.

While below I list several weaknesses that I expect should be fully addressed by the authors, I do believe that the main findings reported in the manuscript are generally well supported. Likewise, the work described in this manuscript, in my opinion, constitutes a substantial and important step forward in our understanding of fungal-host interactions.

1) A major issue is the PL-CL comparison done at 11 days post inoculation throughout the manuscript. It is well established in the field, and also shown in Fig S1, that at d11 post PL infection, no *Candida* cells remain in the tissue (PL infection is cleared ~72h after infection). CL-inoculated animals, on the other hand, have a stable fungal load at d11 which persists for weeks. It is unclear what to make, then, of comparisons like the ones shown in Figs. 1C and 1J-L. As it is, what the authors are actually comparing is the effect of post PL infection to active CL colonization. But that's not how the data are described or the results interpreted. If the purpose is to reveal effects elicited by the CL strain, the control should be 'sham.' If the purpose is to compare PL-CL effects, the comparison should include an early time point for PL because only then there's a clear effect of fungal presence in the tissue. This is appropriately done only for a few cytokines (Figs. 2A-B) but not for the larger scRNA-Seq or epithelial-enriched RNA-Seq included in Fig. 1.

2) Line 109 and Fig. 2A: "Acute infection with the PL *C. albicans* strain led to early TNFalpha and IL-1beta induction." In Fig 2A, the prominent cytokines up only in PL at day 2 seem TNFalpha and IFNgamma (not IL-1beta).

3) Lines 122-124 and Fig. 2G: "CD4 T cells were exclusively found in the epithelial and submucosal layers of commensal colonized mice." The staining in Fig 2G which supports this statement was done with animals 11 days post infection. As stated above in point #1, by this time the PL-infected mice have already cleared the fungus. The statement implies that CD4 T cells are never found during the entire course of infection with PL isolate. But at 2 days post infection, CD4 T cells may well be found in the PL-infected tissue.

4) Fig 4. While the transitory nature of the epithelial K14 expansion seems clear, the connection of this phenomenon to "Candida-specific immunity" seems vague. The increase in Ca-specific IL-17 secreting cells (Fig. 4D) observed at similar timepoints is a correlation but does not indicate causation. The data shown in Rag1^{-/-} animals, while consistent with the author's premise, is incomplete (e.g. we don't know if the epithelial expansion takes place in these animals upon CL colonization) and not very specific (the general lack of mature B/T cells in these animals likely have myriad effects). At the very least, the authors should provide a more nuanced interpretation of the data.

Version 1:

Reviewer comments:

Reviewer #1

(Remarks to the Author)

Although the authors revised the manuscript, I found several criticisms that still make the manuscript unsuitable for publication.

Here below some of the concerns highlighted in the first round:

1. The main limitation of the manuscript is related to the definition of the two strains as 'commensal' and 'pathogenic'.

1. The authors responded to this criticism by citing previously published manuscripts related to strain 101 and SC5314. The cited publications investigated how the two strains are virulent on mouse models of oral candidiasis. These papers suggest that the two strains differentially activate the host response, although Th17 differentiation was indistinguishable after day 7 post-infections.

They also claim that the response (delayed) in the host may be a sign of commensalism.

But, the paper here revised is the first where the strains are defined as 'commensal' or 'pathogenic-like' strains, probably to oversimplify the already published papers. Therefore, I don't see the point of not leaving the original names of the strains like they are SC5314- and 101-strains, and as already defined by previous publications.

2. Figure S1 A is very surprising. Authors claim that the CL strain is not inducing inflammation but a non-inflammatory pathogen can't be associated with such a high, persistent mouth colonization. Data should be presented by showing the standard deviation and if there is also dissemination in other tissues.
2. Here, the authors reply by using data and observations collected by already published papers without really supporting their hypothesis. The 'persistent' colonization in a model of oral infection can't be interpreted as commensalism. Indeed, already published papers claim that the 101 strain is inducing a delayed response, which may only underline a much higher virulence.
- Plus, in Fig. S1B, mice infected with 101 strains start to clear the infection, as do the SC5314-infected mice. Again, I don't see the point at all in defining this experiment as a commensalism model if mice slowly clear the infection.
3. Authors have to clarify or modify the model, by considering also that *Candida albicans* is a human gut commensal and not properly a mouse commensal.
3. Authors did not clarify why they decide to chose a model of oral mouse infection to study human commensalism. A more realistic approach would be to compare human isolated strain responses in culture with the human oral mucosa by building 3D cultures.
4. It is not clear why in Figure 1E it is only shown the histology of mice infected with the CL strain and not the PL strain. Same issue for figure 1F, 1G, 1M.
4. Although I appreciated the data added in Fig.S3, I still would suggest putting the histology of the two strains in the same figure in order to appreciate the underlined differences. To show a quantitative objective increase of epithelial expansion, it is required to have higher-resolution images with a quantification score obtained by scanning the whole mouse tongue. The whole scanned picture should be added. In Fig. 1 the histology at D11 is taken from a different portion of the mouse tongue compared to naïve, D2, D5, D8.
5. In figure 2, authors claim that CL is not inducing inflammation. This sentence is profoundly wrong, considering the pathogenic colonization overtime that the CL strain is inducing and also the evidence that the CL strain is inducing IL-17 and IL-22, two pro-inflammatory cytokines. Data coming from the assay in figure 2A should be repeated eventually by using conventional ELISA, since it is extremely surprising that IL-17 is induced and there are not T cell polarizing cytokines. In other words, do the author know how CD4+IL-22+ T cells were polarized if no inflammation was induced?
- Figure R1 should be transferred to the paper.
 - In Figure R1 authors should put the other time points of the infection.
 - The definition: 'classical cytokine' is not scientifically recognized in the immunological field. To which group of cytokines do the authors refer?
 - The authors did not explain or give any hypothesis on the unexpected IL-22 expansion in the 101 infection model without having polarizing cytokines;
6. Figure 2H is showing plots not clear. Gating strategy is not presented. CD4+IL-22+T cells should be shown.
6. Fig.2H does not show Th17 cells. Unfortunately, the plot only represents a poor staining. The oral mucosa is full of CD4+T cells, which are a well distinct population. Here it is scarcely represented. A major quality of cell staining should be shown.
7. Figure 4H is not clear. Why is keratin still there with no T cells?
7. This is a very important point. It is not clear how the first half of the paper supports the idea that T cells are responding to 101 strain and are fully responsible for IL-22 release, but in Rag1^{-/-}, this response is still there. It is not clarifying the concept to state that this is a compensatory response, since they need to understand if other cells (most probably ILCs) are involved, together with T cells, in responding to the 101 strain. It might be the case to understand if the T memory resident (already shown by previous manuscripts) cells are affecting the IL-22 production.

Reviewer #2

(Remarks to the Author)

The authors have done extensive work to revise the manuscript and have addressed my questions.

Minor point: The persistence of IL-22 signal after CD4⁺ T-cell depletion suggests there are additional IL-22-producing cells in this model. It would further strengthen the manuscript, though not be strictly required, to characterize these other sources using the IL22-TdTomato reporter mouse (for example, by FACS or imaging analysis of non-CD4⁺ populations).

Reviewer #3

(Remarks to the Author)

The authors have satisfactorily addressed all the points I had raised. I believe the manuscript is ready for publication.

Version 2:

Reviewer comments:

Reviewer #1

(Remarks to the Author)

I sincerely thank the authors for the review but unfortunately, I am feeling very uncomfortable because my vision is profoundly far from the authors and also from the other reviewers. I would only like to share here the content of my view:

- My concerns are not only related to 'terminology': commensal/pathogenic. My concerns are absolutely science-based. In my view, the authors use a simple model of oral candidiasis challenging mice with two different strains, but in my opinion, this is a model of local infection and not a commensal model.
- The comparison between CL and PL strains is not often shown without any particular reason. Certain results are comparing the two strains; others are not. This is not acceptable.
- I was asking for a better quantification of histologic analysis between the two strains, but this was not accomplished. The scanning technology on tissues is not that impossible, and I am sure that if you want to demonstrate how commensals change the tongues, you may achieve such a result.
- I was asking for a better representation of CD4+T cells producing IL-22 by FACS (which is the main focus of the paper) than the poor quality single plot represented in Figure 2h, but this was not modified.
- The confused interpretation of results using Rag ko mice is still present since the contribution of ILC3 in the first part of the paper is not addressed, and not even mentioned.
- The last figure is hard to understand and contributes to confusing messages.
- They claim the CL strain induces Th22, but no single hypothesis of a possible polarizing cytokine is exposed, and no one cytokine is measured, proving a Th22 polarization.
- To conclude, I think the authors are only showing that certain strains induce IL-22 just because the CL strain does not produce hyphal invasion compared to the PL strain. Indeed, after 30 days, colonization and keratinization are gone and this is because the model only shows a superficial infection.
- The production of IL-22 and the high K14 expression CAN'T BE SEEN as commensalism; this is very dangerous. We, immunologists, know very well that HIGH IL-22 may represent a chronic stimulation of cancer progression (nature immunology Article <https://doi.org/10.1038/s41590-025-02149-z>), this is also the reason why Candida is well under investigation in the cancer microenvironment.

Reviewer #2

(Remarks to the Author)

The authors have fully addressed my previous concerns

Version 3:

Reviewer comments:

Reviewer #2

(Remarks to the Author)

*Editorial note: This reviewer was asked to comment in the absence of reviewer 1 on the authors response to the prior concerns raised by reviewer 1.

Reviewer 1's main critique appears to focus on the conceptual framework of defining "commensal" versus "pathogenic" behavior using two distinct Candida strains, rather than assessing a continuous spectrum of infection outcomes within a single strain. In practice, such a continuous infection-gradient experiment may not be readily feasible at this stage. I raised a related point in my initial comments, particularly regarding interpretation across early versus late phases of infection.

That said, the authors have made a substantial effort to strengthen both the manuscript and their rebuttal. The additional experiments performed in response to Reviewer 1's questions are, in my view, adequate and address the major concerns. Importantly, the study limitations are appropriately acknowledged and discussed, which supports a scientifically sound interpretation of the findings.

REVISION#1

RESPONSE TO REVIEWER COMMENTS

We would like to thank the reviewers for the critical reading and constructive comments that helped us to improve the manuscript. We have addressed all comments and provide a detailed point-to-point reply to all comments below with changes shown in yellow highlight in the revised manuscript.

Reviewer #1 (Remarks to the Author):

The manuscript by Millet et al. describes the immune response of mice infected orally with two different strains of Candida albicans.

The immune response and the activation of the epithelial barrier were investigated. The main highlighted response is the epithelial remodeling mediated by K14 expansion and the IL-22 production, particularly with the 101 Candida albicans strain. The manuscript is well written, and the topic is clinically relevant, but I found essential conceptual and technical limitations that needed to be carefully clarified.

1. The main limitation of the manuscript is related to the definition of the two strains as 'commensal' and 'pathogenic'. Conceptually and scientifically, it is profoundly misleading and it needs to be explained or demonstrated in a completely different way. The oral model used by the authors is a simple mouse model of an acute infection, it is wrong to consider it as a 'commensalism' model. The infection is persistent with the 101 strain and this is not identified as a 'symbiotic' relation with the host but conversely as an opportunistic behaviour.

Response: We appreciate the reviewer's insightful comment regarding the use of the terms "commensal" and "pathogenic" to describe the two *C. albicans* strains in our study. A commensal microbe is one that induces no perceptible or only clinically inapparent damage to the host, often established early in life and resulting in a stable host-microbe relationship without disease (PMCID: PMC97744). Thus, labeling the *C. albicans* strains as "commensal" or "pathogenic" reflects measurable outcomes of their interaction with the host, focusing on host damage and inflammatory responses rather than simplistic presence or persistence. Furthermore, the CA101 (CL) strain's persistent colonization with curbed inflammation aligns with features interpreted as commensalism, while SC5314 (PL) trigger inflammatory infection and clearance (PMID 28176789; 30873177). We agree that these classifications can oversimplify the dynamic and context-dependent nature of host-microbe interactions, particularly in the case of opportunistic fungi. Thus, we labeled these strains as "pathogenic-like (PL)" and "commensal-like (CL)" to reflect published (PMID 28176789; 30873177; 32719409; 35404986) and observed phenotypes and host responses (this manuscript).

2. Figure S1 A is very surprising. Authors claim that the CL strain is not inducing inflammation but a non-inflammatory pathogen can't be associated with such a high persistent mouth colonization. Data should be presented by showing the standard deviation and if there is also dissemination in other tissues.

Response: We appreciate the reviewer's comment. As discussed in comment 1 the CL strain used (CA101) has been described initially to fail to induce an acute inflammatory response in the

host (PMID 28176789). This phenotype of strain CA101 is well established (PMID 30873177; 32719409), and our new data strengthens the initial observation by the LeibundGut-Landmann lab.

The standard deviation is included, and we added statistics for Fig. S1A. Of note, in the immunocompetent mouse model of OPC (used throughout our study), oral *Candida* infection does not lead to organ dissemination. However, dissemination has been observed during immunosuppression e.g. by using cortisone acetate or mice with defects in barrier immunity (PMID: 24442441; 19204111; 24980544). While oral infection in immunocompetent mice does not lead to organ dissemination, oral infection leads to GI tract colonization (Fig. S1B) in which the PL strain is cleared (as described in PMID: 34724818) but the CL strain remained stable throughout the course of our study.

Figure S1 The commensal-like (CL) isolate CA101 persists in the oral cavity and GI tract following oral infection. **A** Oral fungal burden of wild-type mice infected with indicated strains. Results are median \pm standard error of the mean (SEM) of four independent experiments ($N = 5$ /group). **B** Fecal burden of wild-type mice after oral infection with indicated strains. The y-axis is set at the limit of detection (20 CFU/g tissue). PL, pathogen-like; CL, commensal-like.

3. Authors have to clarify or modify the model, by considering also that *Candida albicans* is a human gut commensal and not properly a mouse commensal.

Response: We have revised the introduction to emphasize that *C. albicans* colonizes different mucosal surfaces. "As a central constituent of the mycobiome, the fungus *Candida albicans* colonizes the oral and gastrointestinal (GI) mucosa of up to 75% of healthy individuals."

4. It is not clear why in figure 1E it is only shown the histology of mice infected with the CL strain and not the PL strain. Same issue for figure 1F, 1G, 1M.

Response: We thank the reviewer bringing this to our attention. We have added histology (PAS staining; S3) and K14 staining for PL infected mice for day 2, 5, 8, and d11 (Fig. S6). The data shows that no epithelial expansion or epithelial remodeling occurs during infection with a PL strain or after infection resolution.

Figure S3. Representative pictures of PAS staining of PL-infected mice over time. N=4-6.

A

B

Figure S6 Time course studies of K14 epithelial expansion.

Representative immunofluorescence pictures of keratin 14 (K14) after indicated time points. Mice were infected with CL (A) or PL strain (B). Scale bar 50 μ m.

5. In figure 1C it is surprising that single cell is not focused on Th17 cells but only Tregs are specifically stained, although Tregs were not the focus of the study.

Response: While scRNA-seq is excellent for measuring RNA transcript levels in individual cells, it may not directly reflect the amount or activity of the corresponding protein (PMID: 33414677). Thus, we determined IL-22 positive immune cell populations (Fig. 2f), as well as the CD4 subpopulations Th17 and Th22 (Fig. 2i) via flow cytometry.

6. Statistics is lacking in figure 1C

Response: This dataset served as an unbiased approach to identify broad cell populations in the oral cavity. Verification of cell numbers and percentages were done separately (Fig. 1g,1l,1m, 2f, 2i) to confirm this dataset.

7. In figure 2, authors claim that CL is not inducing inflammation. This sentence is profoundly wrong, considering the pathogenic colonization overtime that the CL strain is inducing and also the evidence that the CL strain is inducing IL-17 and IL-22, two pro-inflammatory cytokines. Data coming from the assay in figure 2A should be repeated eventually by using conventional ELISA, since it is extremely surprising that IL-17 is induced and there are not T cell polarizing cytokines. In other words, do the author know how CD4+IL-22+ T cells were polarized if no inflammation was induced?

Response: We acknowledge the reviewer's concern and agree that this statement may have been overly broad. Our intention was to convey that, relative to classical inflammatory stimuli, the CL strain does not trigger a robust or classical pro-inflammatory cytokine milieu in the oral mucosa such as the PL strain. However, we recognize that the induction of IL-17 and IL-22 reflects a degree of mucosal immune activation. We have revised the text to more accurately state that "Thus, CL colonization induces limited inflammation characterized primarily by a type 17 response (IL-17/IL-22), rather than broad pro-inflammatory activation as seen during the early phase of PL infection." (line 123ff) The reviewer raises an important point about the cytokine milieu. While we observed significant induction of IL-17 and IL-22, the levels of classical upstream polarizing cytokines (e.g., IL-1 β) were relatively low or unchanged compared to Sham by the multiplex assay. This discrepancy might have been due to sensitivity limitations of the Luminex assay used or to the transient and localized expression of these cytokines. To address this concern, we have repeated the assay using conventional ELISAs for TNF α and IL-1 β on collected samples (included in Rebuttal only; Fig. R1). These results confirmed low levels in the CL-colonized mucosa, supporting the notion that a limited inflammatory environment is sufficient to drive type 17 polarization.

Figure R1 Correlation of TNF α and IL-1 β measured by Luminex and ELISA after 2 days of infection.

8. Figure 2 E is not showing statistics.

Response: We added the statistics performed a Mann-Whitney test.

9. Figure F is showing a wrong statistic test.

Response: We thank the reviewer to bring this to our attention. Indeed, in the original manuscript we did not correct for multiple comparison. In the revised version, we performed Ordinary one-way ANOVA corrected for multiple comparison.

10. Figure H is showing plots not clear. Gating strategy is not presented. $CD4+IL-22+T$ cells should be shown.

Response: We thank the reviewer for this comment. The gating strategy is presented in figure S25. To clarify, Th17 cells were identified as singlets $CD4+ IL-17A+ IL22+$. Th22 cells were identified as singlets $CD4+ IL-17A- IL22+$. We revised the plot graph to indicate this.

Figure 2h Representative flow cytometry plots for intracellular IL-17A and IL-22 levels in CD4+ cells. Th17 cells were identified as CD4+ IL-17A+ IL-22+. Th22 cells were identified as CD4+ IL-17A- IL-22+.

11. Figure 4H is not clear. Why keratin is still there with no T cells?

Response: We thank the reviewer for this comment. While cytokine expression exhibits a significant degree of cell-type specificity, compensation by other immune cell types in the absence of a particular immune cell can occur due to the plasticity and redundancy within the immune system. IL-22 expression is not limited to CD4 T cells. Activated NK cells, NK T cells, CD8 T cells, $\gamma\delta$ T cells, and innate lymphoid cells (ILCs) can also express IL-22 (PMID: 15308104, 17187052, 19100701, 24448096). Importantly, CD4 T cells are sufficient to reduce IL-22-dependent ILC responses, while ILCs proliferate in the absence of mature CD4 T cells (Rag1 KO mice; PMID 24448096). In this line, ILCs are required to control fungal outgrowth during acute OPC (PMID: 23255360). Thus, the immune system demonstrates functional redundancy, where various cytokines and immune cells can perform overlapping roles to maintain immune functionality. This redundancy allows other immune cell types to partially compensate for the absence of a specific immune cell type by producing cytokines otherwise handled by the missing cell type. We took advantage of this system to analyze if *Candida*-specific T cell immunity is required to reverse epithelial remodeling at later stages of colonization. First, we tested if Rag1 KO mice show increased IL-22 levels at the onset of CL colonization (Fig. S21). This data underlines the importance of the immune system's plasticity and compensatory actions. Additionally, we added fungal burden data as well as K14 thickness of Rag1 KO mice after 11 days of CL colonization (Fig. 4f; S21). Of note, the increase in oral burden in Rag1 KO mice is consistent with published data of increased susceptibility of these mice to CL infection (PMID: 32719409). Thus, in the absence of T cells in Rag1 KO, fungal persistent colonization induces upregulation of IL-22 to induce epithelial proliferation and modulation.

f

Figure 4f Oral fungal burden of WT and *Rag1*^{-/-} mice colonized with CL after 11 and 31 days . N=4-5; combined data of three independent experiments. Two-tailed Mann-Whitney Test. The y-axis is set at the limit of detection (20 CFU/g tissue).

Figure S21 Figures S17 *Rag1*^{-/-} mice express IL-22 in the oral mucosa during CL colonization. **A** IL-22 levels in CL-, and Sham-infected mice; *N* = 4. Ordinary one-way ANOVA corrected for multiple comparison. **B** Representative pictures of K14 staining of Sham- and CL-infected mice after 11 days. Scale bar 50 μm. **C** Quantification of K14 thickness in Sham- and CL-infected WT and *Rag1*^{-/-} mice at day 11. *N* = 4. Two-tailed Mann–Whitney Test.

12. Conclusions are in apparent contradiction with the main thesis. If you consider 101 a 'commensal' strain, how do you consider essential for survival the induction of IL-17 and IL-22 and keratinization? This is only suggesting an host essential (and already shown) response against a fungal pathogen.

Response: We appreciate the reviewer's thoughtful question. While IL-17, IL-22, and epithelial remodeling are often associated with responses to pathogens, these mechanisms also play essential roles in homeostatic immune surveillance at barrier sites. In our model, the CL strain induces a non-pathological, calibrated response that promotes barrier integrity and prevents fungal overgrowth, especially in early life. We do not interpret the involvement of IL-17, IL-22, or keratinization as evidence of pathogenicity, but rather as components of a protective adaptation that maintains host-microbe balance without causing tissue damage. Consistent with this interpretation, we identify a novel host-specific mechanism triggered by persistent oral *C. albicans* colonization, in which IL-22 signaling promotes barrier protection without inducing inflammation or tissue damage. Additionally, to the best of our knowledge this is the first report of oral epithelial expansion mediated by non-canonical IL22 receptor complex signaling to amplify antifungal immunity.

Reviewer #2 (Remarks to the Author):

The manuscript by Millet et al. demonstrates that non-canonical IL-22 receptor complex (IL-22RA1/IL-10Rb/gp130) mediates commensal Candida-induced remodeling of the oral epithelium, facilitating immunosurveillance of the oral mucosa. Through comparison of pathogenic and commensal strains of Candida, the authors demonstrate that IL-22 is persistently upregulated until day 11, a process critical for inducing K14+ epithelial thickening via the IL-22RA1/IL-10Rb/gp130 complex, as supported by complementary knockout and inhibition experiments. This study provides valuable insights into non-canonical IL-22R signaling in antifungal responses, but

certain aspects require further clarification to strengthen the conclusions. Below are my comments and suggestions.

1. Figures 2A and 2B indicate that IL-22 expression persists in the CL group but not in the PL group, with significantly higher levels at day 11 compared to IL-17A. However, Figure 2I shows that Th17 cells dominate the CD4⁺ T cell population, with an elevated frequency in the CL group, except for a single low outlier, which likely affects statistical significance. Meanwhile, Th22 cells are relatively low in frequency despite being significantly upregulated in the CL group. To better assess Th22 contributions, I recommend quantifying absolute cell numbers in addition to percentages of CD4⁺ T cells. Another possible explanation is that the Th17 population in Figure 2H might include IL-22/IL-17A double positive cells, which the authors should clarify or emphasize further.

Response: We thank the reviewer for bringing this to our attention. We added the total numbers of Th17 and Th22 cells. In the original manuscript, we identified IL17A⁺ IL22⁺ cells as Th17 cells. We now clarified this in the graph. Please see also comment 11 from reviewer 1.

2. Schönherr et al. (*Mucosal Immunology*, 2017) previously reported that IL-17 is critical for responses to both pathogenic and commensal *Candida* strains in oral infection models, with fungal burden increasing in both *Il17ra* and *Il17rc* knockout mice for both SC5314 and CA101 strains. This finding appears to conflict with the results shown in Figure 2D, where no fungal load differences were observed after IL-17 neutralization. The authors should discuss or speculate on why these observations differ.

Response: We thank the reviewer for this insightful comment. The discrepancy between the conserved requirement of IL-17 signaling for fungal control in IL-17 receptor knockout mice reported by Schönherr et al. and the lack of fungal load differences after IL-17 neutralization (this manuscript) can be understood by considering several factors related to experimental design, timing, and the nature of genetic knockout of a receptor versus neutralization of a cytokine. Schönherr et al. demonstrated that IL-17 signaling through IL-17RA and IL-17RC receptors is essential for controlling fungal burdens of both highly virulent (SC5314) and commensal-like (CA101) *C. albicans* strains in oral infection model. In our manuscript, we depleted IL-17A at the onset of colonization after 11 days of oral colonization. This observed difference in IL-17 requirement would suggest that the effects of IL-17 signaling may be more critical at earlier stages of infection, without immediately affecting fungal burden at the onset of oral colonization as we have analyzed. We revised this in the manuscript line 127 "This suggested that IL-22 may have a prominent role in controlling fungal outgrowth after mucosal colonization, while IL-17 signaling may be more critical at earlier stages of infection, without immediately affecting fungal burden at the onset of oral colonization." Notably, while IL-17A signals through the receptor IL-17RA/RC, IL-17F signaling is also mediated by this receptor complex (PMID: 19575028). Thus, Schönherr et al. inhibited IL-17A/F signaling via genetic knockout of a receptor, while we neutralized the cytokine IL-17A without targeting IL-17F at the onset of colonization.

3. Based on the fungal CFU and K14⁺ epithelial thickness data in Figure 3, thickening appears to be mediated primarily by non-canonical IL-22 receptor signaling through gp130. However, gp130 inhibition only partially impairs antifungal activity (Figure 3M), as CFU levels with gp130 inhibition are similar to those in *Il22ra1* KO (Figure 3E), while K14⁺ thickening persists in the latter. This

raises questions about the relationship between K14+ thickening and antifungal capacity. Could the authors elaborate on how these two phenomena are interconnected in the CL group?

Response: We thank the reviewer for this insightful comment. We added epithelial RNA-seq data in which we show that mice lacking IL-22 or inhibition of gp130 signaling during CL colonization reduced expression of key antimicrobial peptides (Fig. 3q, S20), known to inhibit fungal proliferation (PMID: 24951441.) This data show that epithelial expansion enhances antifungal immunity by amplifying expression of antimicrobial peptides.

Figure 3q Z-scores of antimicrobial peptide genes in epithelial-enriched tissues in WT and *Il22*^{-/-} mice, and WT mice treated with gp130 inhibitor (igp130) after CL infection on day 11. N = 3.

Figure S20 Epithelial thickness correlates with antimicrobial peptide expression. Correlation of K14 thickness and antimicrobial peptide genes (Defb1, Defb3, Sprr2b). N=3. Correlation was determined by Pearson.

4. The inverse correlation between *Candida*-specific IL-17+ cells and K14+ thickness in wild-type mice at days 11 and 31 is intriguing. The authors hypothesize that epithelial resetting is mediated by Th17 expansion, controlling CL *Candida*, while thickening persists in *Rag1*^{-/-} mice. Since CFU levels in wild-type mice are comparable between days 11 and 31, have the authors investigated CFU levels in *Rag1*^{-/-} mice at day 11? Additionally, given the *IL22TdTomato* reporter data indicating CD4+ T cells as the primary IL-22 producers during CL colonization (Fig. 2F), what factors might drive epithelial thickening in the absence of T cells in *Rag1*^{-/-} mice which lack IL-22 producing T cells?

Response: We thank the reviewer for these comments. Please see our reply to comment #11 from reviewer 1 in which we describe that in the absence of T cells in *Rag1* KO, fungal persistent colonization induces upregulation IL-22 to induce epithelial proliferation and modulation. Furthermore, we added data showing that Cd4 KO mice are more susceptible to infection, while

producing IL-22 to expand the epithelial tissue. This data underscores that IL-22 expression is not limited to CD4 T cells during CL infection.

Figure S22 $Cd4^{-/-}$ mice are more susceptible to CL infection, express IL-22 and remodel the oral mucosa. Oral fungal burden of indicated mice colonized with CL. $N = 6$; combined data of two independent experiments. Two-tailed Mann–Whitney Test. The y-axis is set at the limit of detection (20 CFU/g tissue). **B** Representative pictures of K14 staining of CL-infected mice after 11 and 31 days. Scale bar 50 μ m. **C** Quantification of K14 thickness in CL-infected mice at indicated time points. $N = 6$; combined data of two independent experiments. Two-tailed Mann–Whitney Test. **D** IL-22 levels in CL-infected mice; $N = 6$. Two-tailed Mann–Whitney Test.

5. The methodology for quantifying invasion depth in Figure 5I&J should be clearly detailed in the Materials and Methods section. Furthermore, the mechanisms underlying the increased fungal invasion observed in IL-22 KO neonates compared to adults remain unclear. Are fungal burdens different between neonates and adults? Have the authors examined the roles of other immune cells in colonization across these age groups? The findings in Figure 5 are primarily descriptive, and more mechanistic insights would strengthen this section.

Response: We have added the description in the methods section (line449) which we published previously (PMID: 28325761). It now reads “To quantify fungal invasion, PAS-stained sections were randomly selected and imaged by light microscopy, and the depth growth of individual fungi relative to surface of the tongue were determined using PROGRES GRYPHAX® software version 1.1.8.153 (Jenoptik)”. For reference, we added a small graph in this rebuttal letter to visualize this (see below). Furthermore, we acknowledge that Figure 5 is primarily descriptive. However, it highlights a critical phenotype- age-dependent susceptibility to fungal invasion in the absence of IL-22 which serves as a foundation for future mechanistic studies. Of note, the experiments are ongoing.

Figure R2 Example of invasion depth measurement.

Minor comments:

1. As all molecular analyses were performed on day 11, when *Candida* is absent in the PL group, how do the authors differentiate the effects of *Candida* presence on epithelial remodeling from baseline mucosal epithelial properties? For example, the lack of CD4⁺ T cells in the submucosal layer of the PL group (Fig. 2G) might simply reflect the absence of *Candida*. Clarification on this point would be helpful.

Response: We thank the reviewer for raising this important point regarding the interpretation of epithelial and immune changes in relation to fungal presence versus baseline mucosal state. We agree that the absence of detectable *Candida* in the PL group at day 11 complicates direct comparisons, as differences in epithelial remodeling and immune cell localization could indeed reflect divergent host responses to *Candida* exposure rather than intrinsic baseline properties. However, our interpretation is supported by several key considerations. Although *Candida* is undetectable by day 11 in the PL group, mice were colonized earlier in the time course, and thus any remodeling or immune changes observed at this time point may represent residual or resolving responses to transient *Candida* exposure. In contrast, the CL group sustains *Candida* colonization through day 11, allowing us to capture mucosal responses under conditions of persistent fungal presence. In some instances, we included uncolonized (naïve or Sham) mice as a reference for baseline epithelial and immune architecture. These controls showed no distinct differences from PL and CL mice in epithelial thickness. However, we agree that the PL phenotype may not fully represent the naïve state, but rather a response to prior *Candida* exposure and subsequent clearance. We added a statement to line 84 "... representing distinct post-infection host states, namely immune and barrier responses post-fungal clearance (PL) and one reflecting ongoing *Candida* presence and interaction (CL)". As the reviewer notes, CD4 T cells are absent from the submucosa in the PL group at day 11, while they are enriched in the CL group. We performed experiments in which we stained for CD4 T cells after 2 days post PL and CL infection. These data indicate that during early oral fungal infection no CD4 T cells migrate in the epithelial and submucosal layers and this absence is independent of the strain (PL vs. CL). We interpret this as a reflection of ongoing antigenic stimulation by *Candida* in CL mice, which likely sustains local CD4 T cell recruitment and retention. To clarify this distinction, we have revised the text in manuscript to "Next, we evaluated the localization of CD4 T cells during fungal persistence within the oral mucosa. CD4 T cells were exclusively found in the epithelial and submucosal layers of CL-infected mice at the onset of colonization, which differed markedly from PL- or Sham-infected mice (Fig. 2g, S8) suggesting that persistent fungal stimulation is required to recruit and retain CD4 T cells in submucosal layers." (line 134).

Figure S8 Representative immunofluorescence pictures of CD4 expression after 2 days of infection. Mice were infected with PL and CL. Scale bar 50µm.

2.The statement "in CL-colonized mice, mucosal IL-22 levels remained high, while IL-17A levels were low at the onset of colonization" (Lines 114–115) should be revised, as IL-17A levels fluctuate significantly between days 2 and 5.

Response: We revised the sentence to “However, in CL-colonized mice, mucosal IL-22 levels remained high, whereas IL-17A levels declined to low levels at the onset of colonization.” (Line 121ff.).

3.The legend for Figure S8 refers to treatment with both isotype and anti-IL-22 antibodies. However, the figure only includes PL and CL groups without showing data for isotype controls. The authors should ensure all experimental controls are represented.

Response: We apologize for providing the wrong figure for the anti-IL-22 depletion experiment. While preparing the manuscript, we accidentally duplicated the K13 distribution of CL and PL infected mice. We have replaced it with the correct data.

Figure S11 Effect of IL-22 depletion on K14 expansion during oral fungal colonization (A) Representative immunofluorescence pictures of keratin 14 (K14) after 11 days of infection. Mice were infected with CL and treated with isotype or α-IL-22 antibody starting day -1 every other day. Scale bar 50µm. **B** Quantification of K14 thickness in indicated mice at day 11. *N*= 6; combined data of two independent experiments. Mann-Whitnev Test.

4.Immunofluorescence images of the control group in Figure S6 would be informative. The aggregated IL-22RA1 signal in the PL group appears less pronounced in the CL group. Is this difference due to prior Candida infection, or is it also observed in uninfected controls?

Response: We thank the reviewer for this comment. We isolated the red channel of the provided pictures showing that no difference in localization or abundance occurs between Sham-and PL-infected mice. However, the IL22RA1 expression increases in amount as well as expand into the upper epithelial layer consistent with our K14 staining in CL-infected mice.

Figure S9 IL-22RA1 distribution during fungal colonization in the oral cavity. A Representative immunofluorescence pictures of IL-22RA1 after 11 days of infection. Mice were infected with PL, CL, and Sham. Scale bar 50 μ m. **B** Red channel IL-22RA1 shown in black.

5. The apparent toxicity or growth inhibition caused by the STAT3 inhibitor (Figure S10C) should be addressed by including a control group treated with the inhibitor alone. This would rule out potential off-target effects or the possibility that STAT3 signaling is essential for epithelial growth or survival.

Response: We thank the reviewer for bringing this to our attention. Indeed, tonic STAT3 activity is required for optimal proliferation of human oral epithelial cells (Fig. S15D). However, the concentrations used in our experiments had no toxic effect on the epithelial cells determined by LDH assay (Fig. S15D). We have added this data to Fig S13D, E and discussed this in line 153.

Figure S13 D Growth curve of human oral epithelial cells in the presence of STAT3 inhibitor. $N=8$. AUC of oral epithelial cells in response to iSTAT3 treatment. Growth was determined by confluency over time. $N=8$. Mann-Whitney Test. **E** Lactate dehydrogenase (LDH) release of iSTAT3 treated oral epithelial cells after 8 hours. Mann-Whitney Test. $N=3$ in duplicate.

6. The function of tocilizumab as an IL-6R-neutralizing antibody should be stated in the text or the legend for Figure S15.

Response: We have added the information to the figure legend. It now reads "Oral epithelial cell proliferation in the presence of IL-6 receptor blocker tocilizumab".

7. The specificity of gp130 signaling in oral epithelium is an interesting observation. Have the authors examined whether similar mechanisms are involved in other mucosal tissues, such as the vaginal epithelium, another major colonization site for *Candida*?

Response: We thank the reviewer for this insightful comment. We agree that the vaginal epithelium represents an important mucosal site for *Candida* colonization and pathogenesis. While our current study focused specifically on the oral mucosa to dissect the unique features of IL-22/gp130 signaling in this context, we recognize the relevance of investigating whether similar non-canonical pathways are operative in the vaginal mucosa. At this stage, we have not yet extended our analyses to the vaginal epithelium. However, we note that prior studies have reported tissue-specific differences in IL-22 receptor expression and downstream signaling cascades, suggesting that the impact of gp130-mediated signaling may vary across mucosal sites. We are actively pursuing follow-up studies to address whether non-canonical IL-22 receptor signaling is conserved or differentially regulated in the vaginal epithelium during *Candida* colonization and infection.

8. It would be interesting to determine how long CL *Candida* can colonize the oral mucosa. In Figure 4A, fungal burden remains comparable between days 11 and 31, while IL-17+ cell numbers significantly increase by day 31, suggesting immune resetting as proposed by the authors. How do the authors integrate this inverse relationship between IL-17+ cells and K14+ thickening in the context of long-term colonization?

Response: We thank the reviewer for this thoughtful comment. While we limited our observations to day 31, the LeibundGut-Landmann lab has shown stable colonization for >60 days, and in some mice >1 year (PMID: 32719409). The observed inverse relationship between *Candida*-specific IL-17 secreting cell expansion and K14 epithelial thickening may reflect an adaptive mucosal remodeling process (Fig. S24). We propose that during early and onset colonization, IL-22-driven non-canonical signaling via gp130 promotes epithelial proliferation to enhance antimicrobial peptide expression. Over time, however, increasing *Candida*-specific immune cell generation (IL-17 secreting cells) signal a shift toward immune containment rather than further tissue remodeling. One possible interpretation is that *Candida*-specific IL-17 responses limit the need for continued epithelial remodeling by providing direct specific antifungal activity and maintaining fungal burden at a stable set point. Thus, the reduction in K14+ thickening by day 31 may represent a homeostatic reset, where epithelial remodeling initiated by IL-22 is gradually downregulated as *Candida*-specific IL-17-driven immunity becomes the dominant mechanism for sustained fungal control. We have included this interpretation in the revised discussion (line 269).

Reviewer #3 (Remarks to the Author):

In this manuscript, Millet et al. report that oral colonization with a fungal component of the human mouth mycobiota elicits remarkable changes in the structure of the oral epithelium in mice.

Specifically, the authors show that the tongue's epithelial layer expands in a process dependent on IL-22 signaling upon colonization with the oral commensal Candida albicans isolate 101, a strain that can persist for weeks/months in the murine tongues. Furthermore, Millet et al. demonstrate that IL-22 signaling depends on a non-canonical receptor complex composed of gp130 coupled to IL-22RA1 and IL-10RB. Finally, using a mouse model of neonatal fungal colonization, the authors propose that IL-22-mediated immunity and tissue remodeling have key roles controlling fungal invasion early in life.

While below I list several weaknesses that I expect should be fully addressed by the authors, I do believe that the main findings reported in the manuscript are generally well supported. Likewise, the work described in this manuscript, in my opinion, constitutes a substantial and important step forward in our understanding of fungal-host interactions.

1) A major issue is the PL-CL comparison done at 11 days post inoculation throughout the manuscript. It is well established in the field, and also shown in Fig S1, that at d11 post PL infection, no Candida cells remain in the tissue (PL infection is cleared ~72h after infection). CL-inoculated animals, on the other hand, have a stable fungal load at d11 which persists for weeks. It is unclear what to make, then, of comparisons like the ones shown in Figs. 1C and 1J-L. As it is, what the authors are actually comparing is the effect of post PL infection to active CL colonization. But that's not how the data are described or the results interpreted. If the purpose is to reveal effects elicited by the CL strain, the control should be 'sham.' If the purpose is to compare PL-CL effects, the comparison should include an early time point for PL because only then there's a clear effect of fungal presence in the tissue. This is appropriately done only for a few cytokines (Figs. 2A-B) but not for the larger scRNA-Seq or epithelial-enriched RNA-Seq included in Fig. 1.

Response: We appreciate the reviewer's thoughtful comments regarding the PL-CL comparisons at day 11 post-inoculation. We agree that by day 11, the PL strain is typically cleared from the tissue, as also shown in our Fig. S1 and previously reported in the literature. Our rationale for including the day 11 comparison was to capture late-phase mucosal remodeling outcomes following exposure to either the PL or CL strain. Specifically, we sought to determine whether prior transient fungal exposure (PL) elicits lasting epithelial or immune changes that are distinct from those maintained by persistent colonization (CL). Thus, this comparison reflects divergent host states- one in which fungal clearance has occurred and another in which the fungus remains stably colonized. However, we agree that this distinction was not sufficiently emphasized in the manuscript text. To avoid confusion, we revised the manuscript (see lines 84; 106; 109; 133; 136) to clarify that the day 11 PL-CL comparison represents distinct post-inoculation host states, one involving immune and barrier responses post-clearance (PL) and one reflecting ongoing fungal presence and interaction (CL). Additionally, we added data for broader comparison, in which we show histology and K14 distribution over the time course if PL infection (Fig. S3; please see comment #4 Reviewer 1), CD4 cell infiltration at early time points for PL and CL infection (day 2; Fig. S8), which allowed us to distinguish changes that are specifically maintained by CL colonization versus those that may have resolved after PL clearance, as well as compare them the uninfected controls. Of note, the comparison of these strains (CL and PL) to naïve mice during early time points to oral infection/colonization has been published (PMID: 30873177) and were confirmed in our studies with our unbiased cytokine measurements in which we show that the PL strain, in contrast to the CL strain, induces a classical proinflammatory response (Fig. 2A).

2) Line 109 and Fig. 2A: “Acute infection with the PL *C. albicans* strain led to early TNFalpha and IL-1beta induction.” In Fig 2A, the prominent cytokines up only in PL at day 2 seem TNFalpha and IFNgamma (not IL-1beta).

Response: We thank to reviewer to bring this to our attention. To highlight the early differences in some classical pro-inflammatory cytokines, we now provide the data in pg/g tissue of IL-1 β , TNF α , and IFN γ in the supplemental material (Fig. S7) and revised the line 116 to “Acute infection with the PL *C. albicans* strain led to early TNF α and IL-1 β induction, whereas the CL *Candida* strain colonized the oral mucosa without inducing classical proinflammatory markers (Fig. 2A; S7).”

Figure S7 Classical proinflammatory cytokines during OPC. Levels of IL-1 β , TNF α and IFN γ in PL-, CL-, and Sham-infected mice over time; N = 5. Ordinary one-way ANOVA.

3) Lines 122-124 and Fig. 2G: “CD4 T cells were exclusively found in the epithelial and submucosal layers of commensal colonized mice.” The staining in Fig 2G which supports this statement was done with animals 11 days post infection. As stated above in point #1, by this time the PL-infected mice have already cleared the fungus. The statement implies that CD4 T cells are never found during the entire course of infection with PL isolate. But at 2 days post infection, CD4 T cells may well be found in the PL-infected tissue.

Response: We thank the reviewer for this comment. We performed experiments in which we stained for CD4 T cells after 2 days post PL and CL infection. These data indicate that during early oral fungal infection no CD4 T cells migrate in the epithelial and submucosal layers and this absence is independent of the strain (PL vs. CL). Please see minor comment 1 Reviewer 2.

4) Fig 4. While the transitory nature of the epithelial K14 expansion seems clear, the connection of this phenomenon to “Candida-specific immunity” seems vague. The increase in Ca-specific IL-17 secreting cells (Fig. 4D) observed at similar timepoints is a correlation but does not indicate causation. The data shown in Rag1 $^{-/-}$ animals, while consistent with the author’s premise, is incomplete (e.g. we don’t know if the epithelial expansion takes place in these animals upon CL colonization) and not very specific (the general lack of mature B/T cells in these animals likely have myriad effects). At the very least, the authors should provide a more nuanced interpretation of the data.

Response: We thank the reviewer for this comment. We now provide data that *Candida*-primed T cells are required and sufficient to initiate the secondary remodeling event within the oral mucosa. CD4 T cells were harvested from WT mice (CL-colonized or Sham-infected 6 weeks after infection) for adoptive transfer into recipient *Rag1*^{-/-} mice. CD4 T cells were injected intravenously into *Rag1*^{-/-} recipient mice 21 day after CL-colonization. After 10 additional days (day 31 post CL-colonization), CFUs and K14 thickness were determined.

Figure 5 **i** Scheme of CD4 T cell adoptive transfer in CL-colonized *Rag1*^{-/-} mice. **j** Oral fungal burden of *Rag1*^{-/-} mice receiving CD4 T cells from Sham-infected (^SCD4 T cells) WT mice and CL-colonized WT mice (^{CL}CD4 T cells). *N* = 4. Two-tailed Mann–Whitney Test. The y-axis is set at the limit of detection (20 CFU/g tissue). **k** Quantification of K14 thickness at day 31. *N* = 4 in duplicate; combined data of two independent experiments. Mann-Whitney Test. **l** Representative pictures of K14 staining of *Rag1*^{-/-} mice receiving ^SCD4 T cells and ^{CL}CD4 T cells). Scale bar 50μm.

REVISION#2

RESPONSE TO REVIEWER COMMENTS (REVISED MANUSCRIPT)

We would like to thank the reviewers for the critical review and constructive comments that helped us to further improve the revised manuscript. We have addressed all comments and provide a detailed point-by-point reply to all comments below with changes highlighted in yellow in the revised manuscript.

Reviewer #1 (Remarks to the Author):

Although the authors revised the manuscript, I found several criticisms that still make the manuscript unsuitable for publication.

Here below some of the concerns highlighted in the first round:

1. The main limitation of the manuscript is related to the definition of the two strains as 'commensal' and 'pathogenic'.

1. The authors responded to this criticism by citing previously published manuscripts related to strain 101 and SC5314. The cited publications investigated how the two strains are virulent on mouse models of oral candidiasis. These papers suggest that the two strains differentially activate the host response, although Th17 differentiation was indistinguishable after day 7 post-infections.

They also claim that the response (delayed) in the host may be a sign of commensalism. But, the paper here revised is the first where the strains are defined as 'commensal' or 'pathogenic-like' strains, probably to oversimplify the already published papers. Therefore, I don't see the point of not leaving the original names of the strains like they are SC5314- and 101-strains, and as already defined by previous publications

Response: We appreciate the reviewer's attention to terminology. As noted in our previous response, a commensal microbe is defined as one that causes no perceptible or only clinically inapparent damage to the host, typically establishing early in life and maintaining a stable host-microbe relationship without disease (PMCID: PMC97744). We use "commensal-like (CL)" and "pathogen-like (PL)" solely as operational descriptors of host responses in the mouse oral model (stated in lines 46 - 52). We now include new data showing that 11 independent (randomly picked) clinical isolates phenocopy CA101 in colonization kinetics and epithelial remodeling (Fig. S7; lines 111 - 113). This demonstrates that the commensal-like category reflects a reproducible host-response pattern within the oral mucosa rather than a single-strain artifact. Our use of "commensal-like" refers specifically to the behavior of these strains during mucosal colonization rather than an inherent loss of pathogenic capacity. [REDACTED]

[REDACTED]

Figure S7 *C. albicans* clinical isolates show CL phenotype. **A** Oral fungal burden of wild-type mice infected with indicated strains. Results are median \pm standard error of the mean (SEM) of a single experiment ($N = 3/\text{group}$). **B** Representative immunofluorescence pictures of keratin 14 (K14) after 11 days of infection with indicated strains. Scale bar 100µm. **C** K14 thickness 11 days post infection with indicated strains. $N = 3/\text{group}$.

2. Figure S1 A is very surprising. Authors claim that the CL strain is not inducing inflammation but a non-inflammatory pathogen can't be associated with such a high, persistent mouth colonization. Data should be presented by showing the standard deviation and if there is also dissemination in other tissues.

2. Here, the authors reply by using data and observations collected by already published papers without really supporting their hypothesis. The 'persistent' colonization in a model of oral infection can't be interpreted as commensalism. Indeed, already published papers claim that the 101 strain is inducing a delayed response, which may only underline a much higher virulence. Plus, in Fig. S1B, mice infected with 101 strains start to clear the infection, as do the SC5314-infected mice. Again, I don't see the point at all in defining this experiment as a commensalism model if mice slowly clear the infection.

Response: We respectfully disagree with the assertion that high colonization alone equates to pathogenicity. Our data show that CL (CA101) colonization does not induce canonical pro-inflammatory cytokines (IL-1 β , TNF α , IFN γ) or T cell-polarizing cytokines (IL-6, IL-23), distinguishing it from acute infection with PL strain (SC5314), while we and others have shown that the CL strain (CA101) does not induce damage (PMID: 28176789, 35404986, 40998980) a hallmark of pathogenicity (PMID: 27027296, 33471869). Furthermore, epithelial remodeling is IL-22-dependent and reversible, suggesting a regulated host response rather than overt pathology. The standard deviation bars were added in the previous version and we clarified the absence of dissemination in other tissues in an immunocompetent host (Fig. S1C; lines 84 -85).

3. Authors have to clarify or modify the model, by considering also that *Candida albicans* is a human gut commensal and not properly a mouse commensal.

3. Authors did not clarify why they decide to chose a model of oral mouse infection to study human commensalism. A more realistic approach would be to compare human isolated strain responses in culture with the human oral mucosa by building 3D cultures.

Response: We respectfully disagree with the assertion that the oral mouse colonization model is inappropriate for studying mucosal fungal interactions and commensalism. While *C. albicans* is indeed a human commensal, the mouse oral mucosa provides a well-established and tractable in vivo system to study host-fungal interactions in the context of an intact immune system and stratified epithelium. This model has been extensively used to investigate both pathogenic and

commensal behaviors of *C. albicans*. Our study builds on this foundation to investigate how persistent colonization induces epithelial remodeling and immune adaptation. The oral mucosa is a highly relevant site for studying these processes, particularly given the stratified epithelial architecture and immune cell composition that closely resemble human tissues (PMID: 40751276). To complement the in vivo findings, we also employed 2D tissue culture models using human oral epithelial cells to dissect IL-22-mediated signaling mechanisms. These experiments confirmed the proliferative effects of IL-22 and the involvement of non-canonical receptor complexes, thereby strengthening the translational relevance of our observations. Thus, our findings provide mechanistic insights into mucosal immunosurveillance that are broadly applicable to human health. Furthermore, current 3D oral mucosa equivalents primarily consist of epithelial and sometimes fibroblast layers. They lack the full repertoire of immune cells, such as tissue-resident macrophages, dendritic cells, innate lymphoid cells, and T cells that are essential for maintaining commensalism and orchestrating antifungal immunity. This complexity is critical because commensalism is not solely an epithelial phenomenon but depends on dynamic immune surveillance and tolerance mechanisms, as we have shown here.

4. It is not clear why in Figure 1E it is only shown the histology of mice infected with the CL strain and not the PL strain. Same issue for figure 1F, 1G, 1M.

4. Although I appreciated the data added in Fig.S3, I still would suggest putting the histology of the two strains in the same figure in order to appreciate the underlined differences. To show a quantitative objective increase of epithelial expansion, it is required to have higher-resolution images with a quantification score obtained by scanning the whole mouse tongue. The whole scanned picture should be added. In Fig. 1 the histology at D11 is taken from a different portion of the mouse tongue compared to naïve, D2, D5, D8.

Response: We appreciate the reviewer's thoughtful comments and suggestions. Our study was specifically designed to investigate the epithelial remodeling induced by the CL strain, which is a central focus of the manuscript. The PL strain, in contrast, does not elicit comparable histopathological changes, and thus was not included in Figures 1E, 1F, 1G, and 1M, which aim to highlight the distinct pathological features associated with CL colonization. We agree that juxtaposing histological images of both strains could enhance the clarity of the observed differences. Accordingly, we have included comparative histology in Figure S3 to provide visual context. However, we respectfully note that our imaging strategy prioritized consistency by analyzing the middle third of longitudinal tongue sections whenever fungal colonization was evident, to ensure consistency across groups and to capture areas most relevant to the mucosal changes under investigation.

5. In figure 2, authors claim that CL is not inducing inflammation. This sentence is profoundly wrong, considering the pathogenic colonization overtime that the CL strain is inducing and also the evidence that the CL strain is inducing IL-17 and IL-22, two pro-inflammatory cytokines. Data coming from the assay in figure 2A should be repeated eventually by using conventional ELISA, since it is extremely surprising that IL-17 is induced and there are not T cell polarizing cytokines. In other words, do the author know how CD4+IL-22+ T cells were polarized if no inflammation was induced?

- Figure R1 should be transferred to the paper.

- In Figure R1 authors should put the other time points of the infection.

- The definition: 'classical cytokine' is not scientifically recognized in the immunological field. To

which group of cytokines do the authors refer?

- The authors did not explain or give any hypothesis on the unexpected IL-22 expansion in the 101 infection model without having polarizing cytokines;

Response: We acknowledge the reviewer's point and have revised the text to clarify that CL colonization induces a limited type 17 response (IL-17/IL-22) without broad pro-inflammatory activation (lines 128 – 129). We have added the ELISA validation (Fig. former R1) to the manuscript (S8B) and clarified that IL-22+ CD4 T cells arise in the absence of classical polarizing cytokines, suggesting alternative pathways of activation (lines 265 -270). The term "classical cytokines" has been removed and replaced with specific cytokines (lines 122 -123).

6. Figure 2H is showing plots not clear. Gating strategy is not presented. CD4+IL-22+T cells should be shown.

6. Fig.2H does not show Th17 cells. Unfortunately, the plot only represents a poor staining. The oral mucosa is full of CD4+T cells, which are a well distinct population. Here it is scarcely represented. A major quality of cell staining should be shown.

Response: We thank the reviewer for this comment. Our flow cytometry gating for IL-17+ and IL-22+ cells was established using a fluorescence-minus-one (FMO) control strategy, where each marker was assessed in the absence of one fluorochrome to define positive populations (Fig. S.26). While we acknowledge that the flow cytometry plots in Fig. 2H are representative, they were intended to illustrate the presence of IL-22–producing CD4⁺ T cell subsets [Th17 (IL17+/IL22+) and Th22 (IL17-/IL22+)] during persistent colonization. These data are supported by multiple independent experiments and are consistent with our reporter mouse data (Fig. 2E–F; gating Fig. S.26), which identify CD4⁺ T cells as the major source of IL-22. Importantly, we would like to clarify that the oral mucosa is not constitutively enriched in CD4⁺ T cells. As shown in Fig. 2G, CD4⁺ T cells are specifically recruited to the epithelial and submucosal layers in response to *Candida* colonization and are largely absent in sham- or PL-infected mice. Thus, the observed CD4⁺ T cell signal reflects a biologically relevant and colonization-dependent infiltration, not a technical limitation.

7. Figure 4H is not clear. Why is keratin still there with no T cells?

7. This is a very important point. It is not clear how the first half of the paper supports the idea that T cells are responding to 101 strain and are fully responsible for IL-22 release, but in Rag1-/-, this response is still there. It is not clarifying the concept to state that this is a compensatory response, since they need to understand if other cells (most probably ILCs) are involved, together with T cells, in responding to the 101 strain. It might be the case to understand if the T memory resident (already shown by previous manuscripts) cells are affecting the IL-22 production.

Response: We thank the reviewer for highlighting this important point. We have clarified the mechanism underlying persistent keratin (K14) expansion in *Rag1*^{-/-} mice and expanded our discussion to address the role of non-T cell sources of IL-22. As shown in Figure 4H and detailed in the Results (lines 207 - 209), *Rag1*^{-/-} mice exhibit sustained K14 expansion during CL colonization despite lacking mature T cells. This occurs because IL-22 production is not exclusively dependent on CD4 T cells. Our new data (Fig. S22D) demonstrate that group 3 innate lymphoid cells (ILC3s) significantly upregulate IL-22 in *Rag1*^{-/-} mice during colonization, compensating for the absence of adaptive immunity and driving epithelial proliferation. Additionally, we discuss this in lines 275 – 278.

Figure S22D % (left), total (middle), and IL-22⁺ ILC3s (right) in CL-infected WT and *Rag1*^{-/-} mice at day 11. *N* = 6. Two-tailed Mann-Whitney Test.

Reviewer #2 (Remarks to the Author):

The authors have done extensive work to revise the manuscript and have addressed my questions.

Response: We thank the reviewer for their positive assessment and for acknowledging our revisions. We appreciate the time and effort invested in reviewing our work and are pleased that the revised manuscript addresses the previous concerns.

Minor point: The persistence of IL-22 signal after CD4⁺ T-cell depletion suggests there are additional IL-22-producing cells in this model. It would further strengthen the manuscript, though not be strictly required, to characterize these other sources using the IL22-TdTomato reporter mouse (for example, by FACS or imaging analysis of non-CD4⁺ populations).

Response: Please see comment 7 from Reviewer #1.

Reviewer #3 (Remarks to the Author):

The authors have satisfactorily addressed all the points I had raised. I believe the manuscript is ready for publication.

Response: We sincerely thank the reviewer for their thoughtful evaluation and constructive feedback. We greatly appreciate the time and effort dedicated to reviewing our work. We are pleased that the revisions have addressed the previous concerns and improved the clarity and impact of the manuscript.

REVISION#3

RESPONSE TO REVIEWERS

RESPONSE TO REVIEWER COMMENTS (REVISED MANUSCRIPT#2)

Reviewer #1 (Remarks to the Author):

I sincerely thank the authors for the review but unfortunately, I am feeling very uncomfortable because my vision is profoundly far from the authors and also from the other reviewers. I would only like to share here the content of my view:

- My concerns are not only related to 'terminology': commensal/pathogenic. My concerns are absolutely science-based. In my view, the authors use a simple model of oral candidiasis challenging mice with two different strains, but in my opinion, this is a model of local infection and not a commensal model.

Response: We acknowledge this concern and have adopted the terminology “persistence-biased” and “clearance-biased” to describe strain-specific tendencies, avoiding the term “commensal-like.” While these strains exhibit classical features of commensal organisms, we agree that the OPC model does not fully replicate commensalism. To address this, we have included a “Limitations of this study” section at the end of the discussion acknowledging this point. However, the OPC mouse model is widely accepted (over 500 publications) for studying mucosal *Candida* interactions *in vivo*. Our study builds on this framework to dissect mechanisms of mucosal immunosurveillance, not to redefine commensalism. Importantly, we have used multiple clinical isolates, all of which consistently exhibit a persistence-biased phenotype, reinforcing that this observation is not limited to a single strain but reflects a broader biological mechanism.

- The comparison between CL and PL strains is not often shown without any particular reason. Certain results are comparing the two strains; others are not. This is not acceptable.

Response: We have included comparisons between strains in previous versions of the manuscript and incorporated additional comparisons based on Reviewer #1's suggestions. To further improve clarity, we will move the relevant data from the supplemental materials into the main figure (Fig. 1e) so that these comparisons are presented clearly and systematically. Comparisons were performed only where necessary to illustrate the main phenotypic differences, such as scRNA-seq, bulk RNA-seq, K14 expression, histology, CFU counts, and sources of IL-22, while minimizing animal use in accordance with ethical guidelines. Using clearance-biased (CB) strains for later-stage analyses would not be informative, as several phenotypes, such as sustained epithelial remodeling and immune compensation, only manifest under conditions of fungal persistence, which occur with persistence-biased (PB) strains.

-I was asking for a better quantification of histologic analysis between the two strains, but this was not accomplished. The scanning technology on tissues is not that impossible, and I am sure that if you want to demonstrate how commensals change the tongues, you may achieve such a result.

Response Quantification of histologic analysis is already presented; however, we understand the desire for whole-tissue imaging. We have now acquired whole-tongue K14 staining images using

a microscope equipped with scanning capabilities and included these in the revised manuscript (Fig. 2f) to provide a comprehensive view of epithelial remodeling.

Figure 2f Representative pictures of K14 staining of whole tongues of Sham, CB- and PB-infected mice after 11 days.

-I was asking for a better representation of CD4⁺T cells producing IL-22 by FACS (which is the main focus of the paper) than the poor quality single plot represented in Figure 2h, but this was not modified.

Response: We misunderstood this point previously. We have now provided a higher-quality FACS image to better represent CD4⁺ T cells producing IL-22 (Fig. 3h). However, we would like to emphasize that the main focus of the manuscript is on epithelial remodeling, not on demonstrating that IL-22 originates from CD4⁺ T cells (Th17/Th22 cells).

-The confused interpretation of results using Rag ko mice is still present since the contribution of ILC3 in the first part of the paper is not addressed, and not even mentioned.

Response: Based on Reviewer #1's queries, we incorporated new data demonstrating IL-22 production by ILC3s in Rag1^{-/-} mice and expanded the discussion to highlight immune redundancy and compensatory pathways. We also want to clarify that we initially did not focus on ILC3s because the IL22 reporter mice did not show differences in TCR-negative cells in the WT setting, suggesting their contribution was not altered under those conditions. However, the contribution of ILC3s is indeed relevant, and was clarified in the revised manuscript's discussion that "While CD4⁺ T cells dominate IL-22 production in immunocompetent hosts, ILC3s compensate in the absence of T cells, sustaining epithelial proliferation in Rag1^{-/-} mice. These findings underscore the critical role of adaptive immunity in terminating epithelial expansion and achieving long-term fungal control." Furthermore, we used Rag1^{-/-} mice to determine whether Candida-specific T cell immunity is necessary for reversing epithelial remodeling during later stages of colonization. Purified CD4⁺ T cells from Candida-colonized or sham-infected wild-type mice were adoptively transferred into Candida-colonized Rag1^{-/-} mice. Adoptive transfer of Candida-primed T cells confirmed that Candida-specific immunity was required to reduce fungal burden and K14⁺ epithelial thickness. This data indicates that Candida-primed CD4⁺ T cells are both required and sufficient to drive secondary epithelial remodeling during colonization.

-The last figure is hard to understand and contributes to confusing messages.

Response: We used neonates because early life represents a critical window for mucosal immune development, where newborns encounter abundant antigenic challenges but have an inexperienced adaptive immune system and reduced innate responsiveness. This immunological context likely explains why oral candidiasis occurs in infants and why compensatory mechanisms, such as IL-22-mediated epithelial remodeling, are essential. Our data show that neonatal oral *Candida* colonization induces K14 epithelial expansion and CD4⁺ T cell infiltration, and that IL-22 deficiency leads to increased fungal burden, loss of epithelial proliferation, all phenotypes observed in adults. However, deeper tissue invasion was not observed in adults. These findings suggest that protective pathways present in adults are underdeveloped or absent in early life, making epithelial remodeling a critical defense mechanism during this stage. While we acknowledge that we do not yet fully understand why remodeling is more pronounced in neonates, our data clearly highlights the importance of IL-22 in neonatal immune defense during fungal encounter.

-They claim the CL strain induces Th22, but no single hypothesis of a possible polarizing cytokine is exposed, and no one cytokine is measured, proving a Th22 polarization.

Response: Based on Reviewer #1's queries, we expanded the discussion to address this point. Our data show that Th22 cells emerge during fungal colonization despite the absence of classical polarizing cytokines, which we interpret as evidence for alternative tissue-specific cues driving IL-22 production. These cues may arise from persistent fungal antigen exposure rather than canonical inflammatory pathways. We have clarified this in the discussion, emphasizing that this finding challenges the conventional paradigm of Th22 differentiation and suggests a non-inflammatory pathway of T cell activation that may be critical for maintaining mucosal homeostasis during commensal colonization. We did not investigate this further since the main focus of the manuscript is epithelial remodeling rather than defining the precise polarization mechanism.

-To conclude, I think the authors are only showing that certain strains induce IL-22 just because the CL strain does not produce hyphal invasion compared to the PL strain. Indeed, after 30 days, colonization and keratinization are gone and this is because the model only shows a superficial infection.

Response: Response: Oropharyngeal candidiasis (OPC) is generally considered a superficial infection of the oral and pharyngeal lining, characterized by *Candida* overgrowth on mucosal surfaces. While typically localized, OPC involves active fungal adaptation and host responses. We respectfully disagree that IL-22 induction is simply due to the persistent strain lacking hyphal invasion. As shown in recent work (Fróis-Martins et al., *Nat Microbiol*, 2025), even persisting strains (CA101 used as we have done) filament and express hyphae-associated genes during colonization. Candidalysin (a fungal toxin) is essential for establishing and sustaining colonization by enabling the fungus to overcome the stratum corneum barrier and resist IL-17-mediated clearance. The resolution of colonization and keratinization after ~30 days reflects restoration of mucosal homeostasis once *Candida*-specific immunity develops, not a limitation of the model. These findings underscore that colonization is an active, dynamic process requiring virulence factors and host responses.

-The production of IL-22 and the high K14 expression CAN'T BE SEEN as commensalism; this is very dangerous. We, immunologists, know very well that HIGH IL-22 may represent a chronic stimulation of cancer progression (nature immunology Article <https://doi.org/10.1038/s41590-025-02149-z>), this is also the reason why Candida is well under investigation in the cancer microenvironment.

Resonse: We agree that high IL-22 levels have been implicated in certain pathological contexts, including cancer progression, as noted in the cited Nature Immunology article. However, our study does not interpret IL-22 production and K14 expansion as indicators of commensalism. Rather, these findings represent protective, transient responses during initial oral Candida colonization, aimed at preventing fungal overgrowth and tissue invasion until Candida-specific immunity develops.

Reviewer #2 (Remarks to the Author):

The authors have fully addressed my previous concerns

Response: We are pleased that the revisions have successfully addressed the previous concerns and enhanced the clarity and overall impact of the manuscript.

REVISION#4

RESPONSE TO REVIEWER COMMENTS

REVIEWERS' COMMENTS

Reviewer #2 (Remarks to the Author):

*Editorial note: This reviewer was asked to comment in the absence of reviewer 1 on the authors response to the prior concerns raised by reviewer 1.

Reviewer 1's main critique appears to focus on the conceptual framework of defining "commensal" versus "pathogenic" behavior using two distinct Candida strains, rather than assessing a continuous spectrum of infection outcomes within a single strain. In practice, such a continuous infection-gradient experiment may not be readily feasible at this stage. I raised a related point in my initial comments, particularly regarding interpretation across early versus late phases of infection.

That said, the authors have made a substantial effort to strengthen both the manuscript and their rebuttal. The additional experiments performed in response to Reviewer 1's questions are, in my view, adequate and address the major concerns. Importantly, the study limitations are appropriately acknowledged and discussed, which supports a scientifically sound interpretation of the findings.

Response: We thank the reviewer for their thoughtful evaluation of the revised manuscript. Based on our responses to Reviewer #1, we appreciate their recognition that the additional experiments and revisions adequately address the major conceptual concerns, and that the study limitations are appropriately acknowledged.